# Hearing Without Noticing? Attention-Aware Stealthy Black-Box Adversarial Audio Attacks

**Tianyi Xu** [1 2] **Cheng'an Wei** [1 2] **Yue Zhao** [1 2] **Kai Chen** [1 2]

## Abstract

Automatic Speech Recognition (ASR) systems, such as those in intelligent assistants, are vulnerable to adversarial examples (AEs). Benign audio clips like music, when embedded with small perturbations, can trick ASR models into recognizing attacker-specified commands. Prior studies focus on minimizing perturbation magnitude to craft AEs. However, they fails to achieve high attack stealthiness against black-box ASR systems in the physical world. In this paper, we introduce the first music carrier selection algorithm and an attention-aware stealthiness loss function to generate stealthy AEs. Extensive evaluations on five commercial ASR APIs and three widely-used voice assistants demonstrate that our method significantly outperforms state-of-the-art techniques in both effectiveness and stealthiness. Notably, in a user study involving 200 participants, 55.6% of participants perceived our physical adversarial examples as benign audio, which is an improvement of over 20% compared to existing methods.

## 1. Introduction

Automatic Speech Recognition (ASR) models have been widely used to recognize users' voice commands in scenarios such as intelligent assistants in cars and smartphones. However, it has been shown that ASR models are vulnerable to adversarial examples (AEs) (Carlini et al., 2016; Alchekov et al., 2023; Du et al., 2020; Li et al., 2019; Vaidya et al., 2015; Zheng et al., 2021). Specifically, attackers generate AEs by superimposing adversarial perturbations onto a carrier audio (e.g., background music) (Chen et al., 2020b;

[1]Institute of Information Engineering, Chinese Academy of Sciences, Beijing, China [2]School of Cyber Security, University of Chinese Academy of Sciences, Beijing, China. Correspondence to: Yue Zhao <zhaoyue@iie.ac.cn>, Kai Chen <chenkai@iie.ac.cn>.

*Proceedings of the 43$^{rd}$ International Conference on Machine Learning*, Seoul, South Korea. PMLR 306, 2026. Copyright 2026 by the author(s).

Yuan et al., 2018; Xia et al., 2023). The attacker's objective is to cause ASR models to misinterpret them as valid commands, thereby enabling device manipulation.

However, achieving real-world over-the-air attacks against black-box commercial ASR models typically requires stronger perturbations, resulting in significant deficiencies in stealthiness. For instance, user studies on Ni-Occam (Zheng et al., 2021) reveal that only 10.78% of volunteers perceived their generated audio as normal, while results from KENKU (Wu et al., 2023) indicate that 46% of participants could identify the embedded commands. Many adversarial audio samples contain perceptible noise or even audible commands, which can easily raise user suspicion.

We observe that existing works (Carlini & Wagner, 2018b; Chen et al., 2020a; Wu et al., 2023) seek to improve stealthiness primarily by reducing perturbation magnitude and balancing the trade-off between robustness and stealthiness. However, our findings indicate that human auditory perception is not merely a physical reception of sound, but also involves selective attention. Consequently, stealthiness is not only a function of perturbation magnitude. It can also be further improved by reducing the extent to which perturbations capture human attention. Achieving this objective requires addressing the following two challenges:

- *Challenge 1: Mathematically modeling stealthiness.* Unlike perturbation magnitude, the notion of an "attention-insensitive" perturbation relies heavily on subjective auditory perception, and there is currently a lack of effective objective mathematical constraints to model it.

- *Challenge 2: Identifying attention-compatible carriers.* For a given perturbation, a music carrier that could mask the perturbation can reduce the auditory attention and improve stealthiness. However, determining computable criteria for selecting music clips that are compatible with varying target commands remains a challenge.

**Our Method.** To address these challenges, we propose the Hearing Without Noticing (HWN) method. First, we formulate a novel attention-aware loss function to enhance the stealthiness of adversarial perturbations. Based on Predictive Coding Theory (Friston, 2010) and Auditory Saliency (Kayser et al., 2005), which indicate that the hu-

man attention is highly sensitive to unexpected sound, we aim to minimize the acoustic anomalies of the perturbation. To mathematically compute the acoustic anomalies, we adapt the Structural-Residual Decomposition (Hou & Zhang, 2007) to extract the spectral features, explicitly suppressing the prominent residual components that trigger auditory attention.

Second, we propose the attention-compatible carrier selection algorithm. Leveraging the human auditory masking effect of psychoacoustics (Zwicker & Fastl, 2013), this algorithm calculates the masking threshold curves of music clips and performs spectrogram alignment against specific target commands to efficiently identify the most suitable music segments to serve as carriers for adversarial attacks.

We evaluated our method on five cloud ASR services in the digital domain (Google (goo, 2025), Microsoft (mic, 2025), Alibaba (ali, 2025), Tencent (ten, 2025), Openai (Radford et al., 2023)) and three mainstream voice assistant applications in the physical domain (Gemini (Gemini Team & Google, 2023), Amazon Alexa (ale, 2025), Doubao (ByteDance, 2025)). In over-the-air physical attacks, our method achieved a 100% attack success rate against all targets, matching the performance of existing state-of-the-art methods with greater stealthiness. Human perception study involving 200 participants showed that at least 55.6% of them could not distinguish our adversarial examples from benign music, whereas this ratio was only 34.5% for existing methods. Finally, defense and ablation experiments demonstrate the superiortiy and rationality of our method.

In summary, our contributions are as follows:

- By exploring a new perspective on stealthiness, we propose an attention-aware stealthy adversarial attack method against black-box ASR systems. Specifically, we enhance stealthiness by designing an attention-compatible adaptive carrier selection algorithm and a novel attention-aware optimization strategy.

- We comprehensively evaluated the effectiveness and stealthiness of the attack on attack experiments and human perception study. The results indicate that the stealthiness of our method is superior to existing mainstream methods, making stealthy attack possible in physical world.

Code and audio demos are available at https://github.com/Spa-rkle/HWN-Attack.

## 2. Background

### 2.1. Adversarial Audio Attacks

Adversarial attacks can be categorized into white-box and black-box attacks. White-box attacks (Carlini et al., 2016; Chen et al., 2020a; Carlini & Wagner, 2018a; Qin et al., 2019; Schönherr et al., 2020; Yakura & Sakuma, 2018; Yuan et al., 2018) require complete knowledge of the target system to allow for direct gradient access. Black-box attacks (Abdullah et al., 2019; Chen et al., 2020b; Du et al., 2020; Li et al., 2019; Vaidya et al., 2015) represent the most challenging yet realistic scenario, where the attacker has no access to internal system information and can only observe the final decision outputs. "NI-Occam" (Zheng et al., 2021) is a non-interactive physical attack targeting voice-controlled devices, which introduces a randomization strategy at the beginning of each optimization iteration and employs the AdaBelief optimizer (Zhuang et al., 2020). Similarly, "KENKU" (Wu et al., 2023) is an attack method that does not rely on a specific target ASR system. Although these methods can successfully attack black-box systems under specific conditions, their imperceptibility has not yet reached an ideal level.

### 2.2. Speech Recognition and Auditory Perception Mechanisms

Modern ASR systems simulate human auditory processing by transforming acoustic signals via STFT and mapping them onto the Mel (Stevens et al., 1937) or Bark scale (Zwicker, 1961). While traditional approaches relied on decorrelated features like MFCCs (Davis & Mermelstein, 1980), state-of-the-art End-to-End models utilize high-resolution **Log Mel spectrograms** (Radford et al., 2023) to capture fine-grained spectral envelope features (Fant, 1960).

Unlike the mathematical processing in ASR models, human perception is governed by complex psychoacoustics (Zwicker & Fastl, 2013) and cognitive mechanisms. A fundamental phenomenon is the **Masking Effect**, where dominant sounds (the masker) elevate the hearing threshold of adjacent weaker signals (the maskee), rendering them inaudible. Beyond these physical limits, perception is driven by cognitive processes such as Predictive Coding (Friston, 2010) and Auditory Saliency (Kayser et al., 2005). The brain acts as a "prediction machine," suppressing predictable, stationary signals (e.g., background noise) while directing attention to acoustic anomalies.

## 3. Motivation and Problem Statement

### 3.1. Motivation: Why Magnitude is Not Enough

While existing black-box attacks against ASR systems predominantly focus on minimizing perturbation magnitude, they often overlook a fundamental principle of human cognition: auditory perception is not merely a physical reception of sound, but a selective attention process. Crucially, a sound can be physically audible ("hearing") yet cognitively ignored ("without noticing") if it fails to trigger the brain's

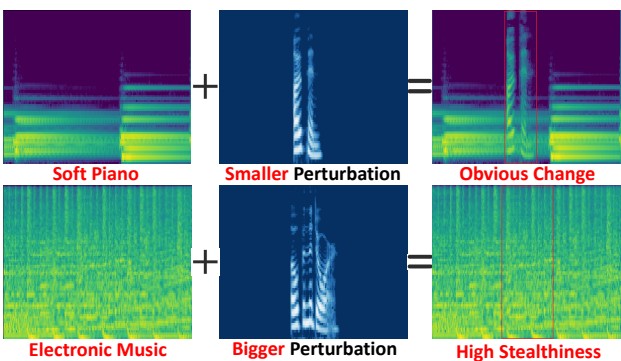

Figure 1. Comparison of spectral artifacts in different acoustic contexts. Sparse carriers (up) amplify the saliency of perturbations, while dense carriers (down) mask them.

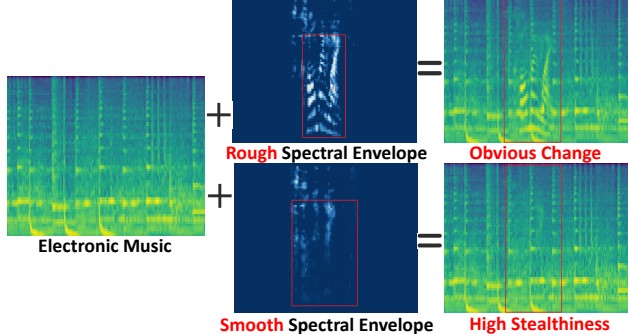

Figure 2. Visualizing different perturbation texture. The smoothed perturbation in Audio 2 (down) is perceptually less salient than the irregular spectral peaks in Audio 1 (up).

bottom-up attention mechanisms. This distinction highlights that physical intensity is a poor proxy for perceptual saliency. Grounded in Predictive Coding Theory (Friston, 2010) and Auditory Saliency (Kayser et al., 2005), the human auditory system acts as a "prediction machine," actively suppressing predictable background streams while directing attention only to acoustic anomalies (e.g., transients and spectral peaks). Driven by this insight, we move beyond simple noise reduction toward attention-aware attack generation, positing that minimizing auditory attention depends on the following two factors.

The first factor is the **acoustic context**. The perceptibility of a perturbation often depends on its acoustic background. As illustrated in Figure 1, in the context of soothing, spectrally sparse piano music (up), even minute perturbations create conspicuous change in the Log Mel spectrogram (consequently, the human ear), as they deviate sharply from the original background spectrogram. In stark contrast, rhythmically dense electronic music (down) can accommodate larger perturbations without triggering attention, as the complex texture effectively masks the noise. This indicates that stealthiness is intrinsically linked to the masking capacity of the carrier.

The second factor is the **spectral envelope** of the perturbation itself. Transient, impulsive noise is characterized by high onset energy and abrupt spectral peaks, which generate high prediction errors in the auditory system and trigger immediate attentional capture. Conversely, statistically stationary broadband noise (like environmental rain or tape hiss) induces a neural habituation effect, fading into the background stream. To validate this, we compared two adversarial samples in Figure 2. Despite using the same music carrier, Audio 2 (down) was rated as more stealthy in human testing. This is because the perturbation in Audio 2 exhibits a smooth, continuous texture in the low-frequency regions, whereas Audio 1 contains irregular spectral peaks (bright regions in the spectrogram).

These observations reveal that minimizing perturbation magnitude does not equate to minimizing attention. To achieve "Hearing Without Noticing," we must utilize the two above factors: (1) **Identifying optimal carriers** that provide high masking capacity, and (2) **Shaping the perturbation texture** to minimize acoustic anomalies. This necessitates a shift from magnitude-based optimization to an attention-aware framework.

### 3.2. Threat Model

Our target encompasses strictly black-box commercial ASR systems, where the attacker has no access to the model architecture, parameters, or training data. For cloud-based APIs: The attacker is restricted to querying the system to obtain the final recognition transcriptions. The objective is to generate successful adversarial examples within a limited, acceptable budget of paid queries. For physical voice assistant applications: The attacker is constrained to performing non-interactive attacks. The generated adversarial examples are played via a loudspeaker in an open space and propagate over the air to nearby speech recognition devices.

We utilize Text-to-Speech (TTS) service to synthesize the target command audio, from which the necessary adversarial features are subsequently extracted.

### 3.3. Adversarial Example Generation

To generate adversarial audio $m_{adv}$ that is perceived as music yet recognized as command $c$, we adopt an optimization-based approach to minimize the deviation in ASR model outputs. In contrast to conventional gradient-based adversarial optimization, our threat model operates in a purely black-box setting and does not rely on costly surrogate models for transfer-based attacks. Consequently, we focus on the critical acoustic features of the target audio rather than the gradients. It is feasible because if the acoustic features of the adversarial example $m_{adv}$ highly match those of $c$, the

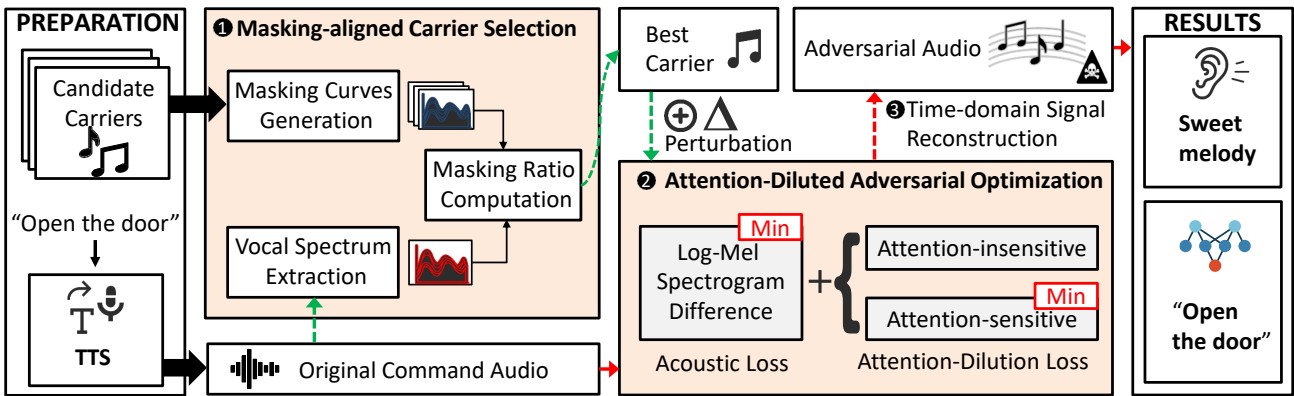

*Figure 3.* Overview of our method.

ASR model is likely to interpret them as the same command. Thus, the objective is redefined as:

$$m_{adv} = \underset{m_{adv}}{\arg\min} \|F(m_{adv}) - F(c)\| \quad (1)$$

where $F(\cdot)$ denotes the acoustic feature extraction function. To ensure the generation of stealthy adversarial audio, stealthiness constraints must be explicitly incorporated into this objective function (as detailed in Section 4.2).

## 4. Attention-Aware Adversarial Audio Generation

Figure 3 illustrates the architecture of the our attack. Given a specific target command text, the process begins by generating the corresponding command audio via a TTS service (Google, 2025). Subsequently, a masking-aligned carrier selection is employed to retrieve an appropriate music segment from candidate carriers to serve as the carrier for the target command. Next, we formulate the attention-diluted adversarial optimization to generate the spectrogram of the adversarial audio. Finally, the adversarial spectrogram is reconstructed in time domain. The final adversarial audio will sound like melody to human but command to ASR model.

### 4.1. Masking-aligned Carrier Selection

As discussed in Section 3.1, for a given perturbation, the selection of carrier yields vastly different perceptual outcomes. The mechanism by which minute perturbations remain concealed within music is intrinsically linked to the auditory **Masking Effect** (Zwicker & Fastl, 2013). When a high-intensity sound occurs at a specific frequency, it elevates the hearing thresholds of adjacent frequencies, implying that surrounding sounds require greater loudness to be perceptible. This phenomenon aligns perfectly with our objective. Consequently, by quantifying the masking capacity of different music carriers, we can identify and select specific

segments capable of masking the largest proportion of the adversarial perturbation. Specifically, the complete algorithmic procedure is outlined in Algorithm 1 and the selection process operates as follows.

We extract fixed-length segments from the candidate music and transform them into spectrograms via STFT. Adhering to the MPEG-1 Psychoacoustic Model (defined in ISO/IEC 11172-3) (Liu & Lee, 1997), we calculate the aggregate masking effects produced within each frequency band of the spectrogram to derive a maximum masking threshold for each band, forming a global masking threshold curve. Since silent frames within the command audio do not require masking, including them in the calculation would bias the selection. Therefore, acknowledging the temporal discontinuity of speech (Sohn et al., 1999), we employ Voice Activity Detection (VAD) (Silero Team, 2025) to strictly filter for the vocal frames containing active speech. Subsequently, we perform the masking ratio computation by comparing the carriers' masking curves with these vocal frames in the target command audio. If the majority of the intensity within a specific frequency band of a command frame falls below the carrier's masking curve, that frame is deemed to be effectively "masked" by the music. Finally, we iterate through candidates to select the carrier segment that masks the maximum number of vocal frames.

### 4.2. Attention-Diluted Adversarial Optimization

According to Section 3.3, the loss function of optimization process can be formulated as follows.

$$\mathcal{L} = \mathcal{L}_{acoustics} + \lambda \times \mathcal{L}_{stealthiness} \quad (2)$$

Specifically, the acoustic feature loss $\mathcal{L}_{acoustics}$ and the stealthiness loss $\mathcal{L}_{stealthiness}$ correspond to the attack robustness and stealthiness of the adversarial audio, respectively. The hyperparameter $\lambda$ governs the trade-off between these two objectives during the optimization process. In practice, we employ a binary search algorithm to efficiently determine the optimal value for $\lambda$. Since both the acoustics

**Algorithm 1** Masking-aligned Carrier Selection Algorithm

---

**Input:** Command audio $C$, Music carrier dataset $\mathcal{D} = \{M_1, \ldots, M_K\}$, Window length $L_{win}$
**Output:** Optimal carrier segment $S_{opt}$

**// Preprocessing**
Calculate power spectrogram $P_C \leftarrow |\text{STFT}(C)|^2$
Get dimensions: $N_{freqs}$ (frequency bins) and $N_{frames}$ (time frames) from $P_C$
Detect vocal mask $V$ by VAD and count vocal frames $N_{vocal}$
Initialize $Score_{max} \leftarrow 0$, $S_{opt} \leftarrow$ None

**for** each music carrier $M_k \in \mathcal{D}$ **do**
    Compute masking threshold matrix $T_k$ using MPEG-1 Psychoacoustic Model
    **for** each candidate start frame $i$ in $M_k$ **do**
        $T_{seg} \leftarrow T_k[\cdot, i : i + N_{frames}]$
        $N_{covered} \leftarrow 0$
        **// Frame-wise Comparison**
        **for** $t = 1$ **to** $N_{frames}$ **do**
            $Count \leftarrow \sum_{f=1}^{N_{freqs}} \mathbb{I}(P_C[f,t] < T_{seg}[f,t])$
            $Ratio_{frame} \leftarrow Count/N_{freqs}$
            **if** $Ratio_{frame} \geq 0.9$ **and** $V[t]$ **then**
                $N_{covered} \leftarrow N_{covered} + 1$
            **end if**
        **end for**
        **// Score Calculation**
        $Score_{curr} \leftarrow N_{covered}/N_{vocal}$
        **if** $Score_{curr} > Score_{max}$ **then**
            $Score_{max} \leftarrow Score_{curr}$
            $S_{opt} \leftarrow M_k[\text{time}(i) : \text{time}(i) + L_{win}]$
        **end if**
    **end for**
**end for**
**return** $S_{opt}$

---

and stealthiness loss function are computed based on spectrograms, we directly introduce the adversarial perturbation $\Delta$ into the spectrogram of the carrier audio, which denoted as $\text{STFT}(m) + \Delta$. $\Delta$ is treated as an optimization variable and is iteratively updated to minimize the loss function.

### 4.2.1. ACOUSTICS LOSS

To ensure a successful adversarial attack, it is fundamental to guarantee that the acoustic features of the AEs are similar to those of the target command audio.

Acoustic features serve as the basis for model recognition, with MFCC and Log Mel spectrograms being the two most prevalent representations. We adopt the latter, which also serves as the standard input feature for the Whisper model

of OpenAI. The advantages of Log Mel spectrograms over MFCC features are shown in Section 5.5.

Specifically, the acoustics loss $\mathcal{L}_{\text{acoustics}}$ in Equation (2) is formulated as follows:

$$\mathcal{L}_{\text{acoustics}} = \text{MAE}\Big( \log(\mathbf{M}_{\text{adv}} + \epsilon), \quad \log(\mathbf{M}_{\text{target}} + \epsilon) \Big) \quad (3)$$

Here, $\mathbf{M}_{\text{adv}}$ and $\mathbf{M}_{\text{target}}$ represent the adversarial and target Mel spectrograms, while $\epsilon = 10^{-10}$ prevents log zero instability. The acoustics loss is defined as the Mean Absolute Error (MAE) between the adversarial audio's Log Mel spectrogram and the target command's Log Mel spectrogram. This MAE metric encourages the adversarial audio to match the target acoustic features while maintaining robustness to small perturbations, as MAE is less sensitive to outliers compared to Mean Squared Error (MSE).

### 4.2.2. ATTENTION-DILUTION LOSS

To improve the stealthiness of our attack, we design the attention-dilution loss to suppress attention-sensitive components of the perturbation.

We formulate this via the **Structural-Residual Decomposition** (Hou & Zhang, 2007) mechanism. According to this mechanism, a signal can be decomposed into a attention-insensitive *structural component* and an attention-sensitive *residual component*. The latter is primarily responsible for triggering attention. Let $\Delta$ denote the adversarial perturbation spectrogram. We employ a Cascaded Multi-Scale Smoothing operator, denoted as $\mathcal{T}(\cdot)$, to extract the structural envelope of the perturbation. The operator $\mathcal{T}$ is defined by sequential convolutions with normalized square kernels $\mathbf{K}_{sq}^{(s)} \in \mathbb{R}^{s \times s}$ and vertical kernels $\mathbf{K}_{ver}^{(s)} \in \mathbb{R}^{s \times 1}$:

$$\mathbf{K}_{sq}^{(s)} = \frac{1}{s^2} \cdot \mathbf{1}_{s \times s}, \quad \mathbf{K}_{ver}^{(s)} = \frac{1}{s} \cdot \mathbf{1}_{s \times 1} \quad (4)$$

where $s$ represents the kernel scale. The structural component $\mathbf{S}_\Delta$ is obtained by $\mathbf{S}_\Delta = \mathcal{T}(\Delta)$, which is the attention-insensitive background. Consequently, the attention-sensitive component, denoted as $\mathbf{R}_\Delta$, is defined as the residual between the raw perturbation and its attention-insensitive component:

$$\mathbf{R}_\Delta = \Delta - \mathbf{S}_\Delta \quad (5)$$

This residual $\mathbf{R}_\Delta$ captures prominent spectral peaks, such as formants and harmonic structures, that induce acoustic anomalies. For best stealthiness, we minimize the energy of these acoustic anomalies using the MSE:

$$\mathcal{L}_{stealth} = \|\mathbf{R}_\Delta\|_2^2 = \frac{1}{N} \sum_{i,j} (\Delta_{i,j} - \mathbf{S}_{\Delta_{i,j}})^2 \quad (6)$$

By minimizing $\mathcal{L}_{stealth}$, we force the adversarial perturbation to conform to a statistically stationary texture, thereby suppressing its attention-sensitive components and evading human attention.

## 4.3. Time Domain Signal Reconstruction

Based on Sections 4.2, we obtain the adversarial spectrogram $\text{STFT}(m) + \Delta$ for a specific target command. Due to the lack of phase information, it is necessary to convert the adversarial spectrogram $\text{STFT}(m) + \Delta$ back into a time-domain audio signal via Griffin-Lim algorithm (Griffin & Lim, 1984), introduced in Appendix E.

For non-interactive scenarios in physical domain (e.g., Gemini), the generation process concludes at this stage. However, for black-box ASR models that support digital iterative queries (e.g., cloud-based API services), we introduce an additional optimization technique to eliminate perceptually insignificant components of the perturbation.

Given the adversarial perturbation $\Delta$, we discretize it into a set of $m \times n$ spectrogram grids $\{\Delta_{i,j} \mid i \in [1, m], j \in [1, n]\}$. Then we define a set of amplitude scaling parameters $\{W_{i,j} \mid i \in [1, m], j \in [1, n]\}$, where each $W_{i,j}$ adjusts the amplitude of the corresponding grid $\Delta_{i,j}$ via the operation $W_{i,j} \cdot \Delta_{i,j}$. To optimize $W$, we employ the Covariance Matrix Adaptation Evolution Strategy (CMA-ES) (Hansen, 2016), recognized as one of the most robust and widely used evolutionary algorithms. The objective function is formulated as follows:

$$\mathcal{L}(W) = \begin{cases} \text{Sum}(|W \cdot \Delta|), & \text{ASR}(\text{iSTFT}(\text{STFT}(m) + \lambda \cdot \Delta)) = y^* \\ +\infty, & \text{otherwise} \end{cases} \tag{7}$$

where $\mathcal{L}(\cdot)$ denotes the objective function, $y^*$ represents the target command, and $\text{Sum}(\cdot)$ measures the magnitude of modification (calculated as the sum of absolute values of the adversarial perturbation). By minimizing the loss, the perturbation which is insignificant to model will be removed by minimizing the loss.

## 5. Experiment

We evaluate our method on various ASR models and conduct a human perception study to assess the stealthiness of adversarial examples (AEs). For our main experiments, we select two highly representative and reproducible state-of-the-art (SOTA) baselines: KENKU (Wu et al., 2023) and Occam & NI-Occam (Zheng et al., 2021). Due to the lack of open-source implementations for more recent methods (ZQ-Attack (Fang et al., 2024) and EvilHarmony (Yuan et al., 2025)), a direct experimental comparison is infeasible. However, we provide a detailed literature comparison with these methods based on their reported results in Appendix B, demonstrating that our approach maintains the leading performance in human perceptual stealthiness while preserving robust practical applicability. Implementation details are shown in Appendix D.

## 5.1. Setup

**Target Commands, TTS and Carriers** Following existing studies (Wu et al., 2023; Zheng et al., 2021), we selected 10 common commands as attack targets. 10 Commands are abbreviated as C1–C10 (C1: call my wife, C2: how is the weather, C3: make it warmer, C4: navigate to my home, C5: open the door, C6: open the website, C7: play music, C8: restart phone, C9: take a picture, C10: turn off the light). Target command audio is synthesized utilizing TTS platforms (Google, 2025; Microsoft, 2026). We investigate the generalization capabilities across diverse voice types in Section 5.7. For the proposed method, 42 distinct music tracks were selected from a music platform. The dataset consists of 27 electronic/rhythmic tracks and 15 sparse piano/ambient sounds, which act as the candidate carriers. For KENKU and Occam, we used the same carriers in the interest of fairness.

**Digital Domain Attack** In the experiments, we targeted the following commercial ASR systems: Google (goo, 2025), Microsoft (mic, 2025), Alibaba (ali, 2025), Tencent (ten, 2025), Openai (Radford et al., 2023). These systems provide APIs that allow users to upload local audio files and receive text transcription results from remote servers, with no data loss or additional noise introduced during the process. The justification for evaluating stealthiness metrics solely on successful AEs is provided in Appendix A.4.

**Physical Domain Attack** To further investigate the effectiveness of physical over-the-air adversarial attacks, we evaluated adversarial examples on mainstream speech recognition applications on smartphones, including Gemini (Gemini Team & Google, 2023), Amazon Alexa (ale, 2025), Doubao (ByteDance, 2025) used by millions globally. All adversarial examples were played through a JBL Clip 3 portable speaker at 80% volume, with the output directed toward a smartphone placed nearby on the same table.

**Defense Methods** We implemented Frequency Band Filtering (Eisenhofer et al., 2021) and Audio Turbulence (Yuan et al., 2018) to test the robustness of the attacks. Since attacks rely on high-frequency components and fine-grained perturbations to remain inaudible, signal processing techniques like specific band filtering and noise injection can destroy these delicate structures, leading to attack failure.

## 5.2. Evaluation Metrics

We evaluate attack effectiveness using Attack Success Rate (SR) and stealthiness using Signal-to-Noise Ratio (SNR), complemented by a human perception study to further validate imperceptibility, consistent with prior works (e.g., (Wu et al., 2023)).

**Attack Success Rate (SR)** Adopting the definition from (Wu et al., 2023; Zheng et al., 2021), an adversar-

*Table 1.* Summary of digital attack performance (Average over 10 commands and 5 APIs). We report SR, human perception results (Normal, Noise, Talk, 1st, 2nd), and quality metrics (MOS, SNR). Best results are **bolded**.

| Method | SR | Normal | Noise | Talking | 1st | 2nd | MOS | SNR |
|---|---|---|---|---|---|---|---|---|
| Occam | 100% | 19.41% | 26.77% | 53.79% | 0.37% | 1.84% | 3.21 | 9.63 |
| Kenku | 100% | 40.72% | 14.06% | 45.20% | 4.81% | 5.18% | 3.71 | 15.66 |
| **Ours** | 100% | **56.25%** | 27.67% | **16.07%** | **0.14%** | **0.63%** | **4.06** | **21.50** |

*Table 2.* Summary of physical attack performance (Average over 3 voice assistants). We report SR, human perception results (Normal, Noise, Talk, 1st, 2nd), and quality metrics (MOS, SNR). Best results are **bolded**.

| Method | Voice Assistant | SR | Normal | Noise | Talking | 1st | 2nd | MOS | SNR |
|---|---|---|---|---|---|---|---|---|---|
| Occam | Gemini | 66.67% | 0.00% | 33.30% | 66.70% | 0.00% | 0.00% | 1.37 | 1.19 |
| | Alexa | 66.67% | 0.00% | 18.50% | 81.50% | 0.00% | 3.70% | 1.33 | 0.86 |
| | Doubao | 100.00% | 0.00% | 55.60% | 44.40% | 0.00% | 0.00% | 1.41 | 3.33 |
| | **Average** | 77.78% | 0.00% | 35.80% | 64.20% | 0.00% | 1.23% | 1.37 | 1.79 |
| KENKU | Gemini | 100.00% | 33.30% | 0.00% | 66.70% | 11.10% | 18.50% | 3.11 | 9.56 |
| | Alexa | 100.00% | 25.90% | 18.50% | 55.60% | 0.00% | 7.40% | 2.56 | 3.91 |
| | Doubao | 100.00% | 44.40% | 48.10% | 7.40% | 0.00% | 0.00% | 3.48 | 9.37 |
| | **Average** | **100.00%** | 34.53% | 22.20% | 43.23% | 3.70% | 8.63% | 3.05 | 7.61 |
| **Ours** | Gemini | 100.00% | 55.60% | 44.40% | 0.00% | 0.00% | 0.00% | 4.45 | 16.22 |
| | Alexa | 100.00% | 29.60% | 43.00% | 27.40% | 0.00% | 0.00% | 3.56 | 15.58 |
| | Doubao | 100.00% | 81.50% | 14.80% | 3.70% | 0.00% | 0.00% | 4.56 | 16.55 |
| | **Average** | **100.00%** | **55.57%** | 34.07% | **10.37%** | **0.00%** | **0.00%** | **4.19** | **16.12** |

*Table 3.* Detailed evaluation results on Google API. 10 Commands (CMD) are abbreviated as C1–C10. Statistical metrics include human perception results (Normal, Noise, Talking), and quality metrics (MOS, SNR).

| CMD | Normal | Noise | Talking | MOS | SNR |
|---|---|---|---|---|---|
| C1 | 60.70% | 28.60% | 10.70% | 4.18 | 24.59 |
| C2 | 96.40% | 3.60% | 0.00% | 4.64 | 21.97 |
| C3 | 92.90% | 3.60% | 3.60% | 4.71 | 24.04 |
| C4 | 64.30% | 28.60% | 7.10% | 4.29 | 23.52 |
| C5 | 89.30% | 0.00% | 10.70% | 4.54 | 24.67 |
| C6 | 39.30% | 50.00% | 10.70% | 3.79 | 17.38 |
| C7 | 75.00% | 21.40% | 3.60% | 4.39 | 25.07 |
| C8 | 78.60% | 17.90% | 3.60% | 4.32 | 17.73 |
| C9 | 92.90% | 7.10% | 0.00% | 4.61 | 20.84 |
| C10 | 85.70% | 10.70% | 3.60% | 4.39 | 17.01 |
| **AVG** | **77.51%** | **17.15%** | **5.36%** | **4.39** | **21.68** |

ial example is considered successful if it is recognized as the target command at least once in three attempts (playing the same sample three times). One adversarial example is considered successful if it achieves correct transcription. SR is defined as the ratio of successful AEs to the total number of AEs in the test set.

**Signal-to-Noise Ratio (SNR)** SNR is defined as the logarithmic ratio of signal (carrier) power to noise (perturbation) power. In Equation (8), $P$ represents the average power:

$$\text{SNR (dB)} = 10 \log_{10} \left( \frac{P_{signal}}{P_{noise}} \right) \qquad (8)$$

**Human Perception Study** To quantitatively evaluate the perceptual stealthiness, we conducted a human perception study including the following metrics:

- **Normal**: The proportion of participants who perceived the audio as benign music without anomalies.
- **Noise**: The proportion of participants who detected environmental noise or artifacts but no speech.
- **Talk**: The proportion of participants who explicitly heard verbal content or speech components.
- **1st and 2nd**: The proportion of participants who successfully recognized the target command after the first and second listening attempts, respectively.
- **Mean opinion score (MOS)**: The average value of scores that participants rate the level of sound quality on a 5-point scale: 5 (Excellent), 4 (Good), 3 (Average), 2 (Fair), and 1 (Poor).

Detailed questionnaires are provided in Appendix F. The study included several adversarial clips and benign music as a baseline. We recruit 200 native English speakers from the platform of Prolific (Pro, 2025), ensuring no hearing or speech impairments. More Statistical details about the human perception study are provided in Appendix C.

### 5.3. Digital Attack Results

With no data distortion introduced in digital domain, we only evaluate successful adversarial examples to ensure a valid stealthiness assessment. Our method demonstrates

*Table 4.* Ablation study demonstrating the effectiveness of each component. "Selection" denotes our carrier selection algorithm; "Loss" denotes our attention-dilution loss (vs. magnitude constraint). Statistical metrics include human perception results (Normal, Noise, Talking), and quality metrics (MOS, SNR).

| Components | | | Evaluation Metrics | | | | | | |
|---|---|---|---|---|---|---|---|---|---|
| Selection | Feature | Loss | Normal | Noise | Talking | 1st | 2nd | MOS | SNR |
| ✓ | Log Mel | - | 9.28% | 41.43% | 49.28% | 3.58% | 2.83% | 2.66 | 11.85 |
| ✓ | MFCC | ✓ | 35.73% | 42.13% | 22.15% | 0.70% | 0.70% | 3.55 | 9.84 |
| - | Log Mel | ✓ | 7.85% | 22.15% | 70.00% | 9.28% | 2.83% | 2.22 | 14.81 |
| ✓ | Log Mel | ✓ | **55.57%** | **34.07%** | **10.37%** | **0.00%** | **0.00%** | **4.19** | **16.12** |

comprehensive superiority over state-of-the-art baselines, achieving an optimal perceptual stealthiness. Among the five commercial APIs evaluated, our method achieves peak performance on Google API. As reported in Table 1, we attain an SNR of 21.68 dB, with 77.51% of participants unable to distinguish the adversarial audio from benign music. Detailed outcomes for the remaining APIs are provided in the Appendix G. Table 1 summarizes the average comparative results against baselines. In terms of human perception, our approach achieve the highest Normal ratio of **56.25%**, surpassing Kenku (40.72%) and Occam (19.41%) by a large margin. Crucially, we minimize the risk of command leakage: only **16.07%** of participants detected speech artifacts ("Talking") in our samples, whereas this ratio exceeds 45% for both baselines. From an objective perspective, our method also maintains the highest SNR of **21.50 dB**, which means smaller perturbations.

### 5.4. Physical Attack Results

Table 2 presents the performance of over-the-air attacks against three commercial voice assistants (Gemini, Amazon Alexa, and Doubao). Overall, our method demonstrates a decisive advantage over state-of-the-art baselines, achieving a perfect Attack Success Rate (SR) while maintaining better perceptual stealthiness. In terms of robustness, our method achieves a **100%** SR, matching the performance of KENKU and significantly outperforming Occam (77.78%). Regarding perceptual stealthiness, we achieve **55.6%** of Normal ratio and a SNR of **16.12** on average, far exceeding KENKU (34.5% / 7.61) and Occam (0.0% / 1.79). The attack effectiveness exhibits variation across different assistants. Notably, against Doubao, our method achieves an impressive Normal ratio of 81.5% with a MOS of 4.56. Even on the challenging Amazon Alexa, our method maintains a low "Talking" ratio of 27.4%, outperforming Kenku and Occam.

### 5.5. Ablation Study

As shown in Table 4, the removal of individual components leads to a collapse in stealthiness performance. Most notably, omitting Carrier Selection results in a 70.00% "Talk" rate, meaning the attack is easily detected as malicious voice commands. The drop in SNR (**16.12** to 9.84) demonstrates

*Table 5.* Robustness evaluation under defense methods. We report the Attack Success Rate (SR) against Frequency Band Filtering and Audio Turbulence. Best performance on both stealthy samples and perceptible samples are **bolded**.

| Method | Type | Filtering | Turbulence |
|---|---|---|---|
| Occam | Stealthy | 0.00% | 0.00% |
| | Perceptible | 30.77% | 23.08% |
| Kenku | Stealthy | 7.69% | 15.38% |
| | Perceptible | **61.54%** | **61.54%** |
| **Ours** | Stealthy | **30.78%** | **38.46%** |
| | Perceptible | 3.85% | 3.85% |

the coarse-grained nature of MFCC features compared to Log Mel features. Similarly, relying on perturbation magnitude constraints reduces the "Normal" perception to only 9.28%. These comparisons indicate that all the components in our method are important.

### 5.6. Defense Results

We evaluated the SR of three methods under Frequency Band Filtering and Audio Turbulence. Our method reaches superior robustness over **stealthy** AEs. We categorize an adversarial example as **stealthy** if less than 50% of human evaluators perceive verbal content (i.e., "talking"); otherwise, it is classified as **perceptible**. Table 5 presents the robustness results under defense mechanisms. Crucially, our method achieves the highest success rate (**30.78% / 38.46%**) specifically among **stealthy** samples. While Kenku exhibits higher robustness on **perceptible** samples, such attacks have no practical utility due to their ease of human detection.

### 5.7. Voice Type Generalization

We evaluated a subset of commands using different TTS service and speaker styles on Openai Whisper. Compared with the baseline in Table 6, our method exhibits robustness across different TTS services and voice styles. Notably, we achieved exceptionally high imperceptibility (**Normat Rate 77.30%**) when optimizing toward a "Woman with happy voice" template. We did observe a relative decrease in performance when using a "man with low voice." The distinct low-frequency characteristics of a male voice are more challenging to mask under relative high-frequency

*Table 6.* Performance comparison across different voice types. The table reports the success rate (SR), perturbation ratios under different conditions (Normal, Noise, Talk), and audio quality metrics (MOS, SNR).

| Voice Type | SR | Normal | Noise | Talking | MOS | SNR |
|---|---|---|---|---|---|---|
| Man with Low Voice | 100% | 36.40% | 45.50% | 45.50% | 3.82 | 15.28 |
| Woman with Happy Voice | 100% | 77.30% | 4.50% | 18.20% | 4.36 | 18.18 |
| Human Recordings | 100% | 50.00% | 13.60% | 36.40% | 3.95 | 13.13 |
| Woman of Google TTS (Baseline) | 100% | 55.57% | 34.07% | 10.37% | 4.19 | 16.12 |

background. Human recordings perform slightly worse, showing an 5.57% decline in Normal Rate, possibly due to device distortion.

### 5.8. Attack Efficiency and Cost

In contrast to prior studies (e.g., Devil's Whisper (Chen et al., 2020b) and Occam (Zheng et al., 2021)) which typically span over 10 hours due to model training and extensive queries, our method drastically reduces the overall generation time to about **50 minutes**. This duration encompasses the entire pipeline, including the masking-aligned carrier selection algorithm which needs to be executed only once. The query overhead and economic cost of our method are also significantly lower. In the digital-domain experiments, the CMA-ES refinement averaged 47 iterations and 707 queries per sample, taking 778 seconds on average. This represents a substantial improvement over Occam, which requires up to 30,000 queries. The average cost for generating an adversarial audio across various commercial APIs is only $0.43. Crucially, in the physical experiments, no interactive queries are applied to the target black-box devices, resulting in a strict **zero-query** attack generation process.

## 6. Discussion

**Characteristics of the Selected Carriers** Analysis of the selected carriers reveals that sparse genres (e.g., piano and ambient music) are rarely chosen by our algorithm due to their limited masking capacity. Instead, the optimal carriers typically exhibit two distinct characteristics:

- *High acoustic density and energy (82.7%).* Sparse music segments fail to establish sufficient masking thresholds to cover phonetic features, making dense electronic or rhythmic tracks highly preferred.

- *Dynamic and complex rhythms (70.9%).* Compared to steady beats, dynamic patterns provide a wider variety of temporal masking windows, which better accommodate the varied structures of human speech.

**Design Choice: Spectrogram vs. Waveform Space** Rather than perturbing the raw audio waveform directly, our attack is formulated in the spectrogram domain based on two considerations:

- *Optimization Landscape.* Our core modules are inherently frequency-domain based. Because magnitude extraction in STFT discards phase information, direct waveform optimization leads to a highly non-convex loss landscape, causing severe gradient oscillation.

- *Phase Preservation.* Music backgrounds contain highly coordinated harmonic structures. Direct perturbation on the raw waveform easily disrupts delicate phase relationships, which introduces audible broadband noise and severely degrades stealthiness.

**Psychoacoustic Mechanism of Residual Suppression** Psychoacoustic studies on Auditory Saliency (Kayser et al., 2005) identify frequency and temporal contrasts as the primary triggers for auditory attention. Our 2D spatial smoothing operation serves as a unified proxy to suppress both. In the time domain, it smooths sudden transients and impulsive clicks to reduce temporal contrast. In the frequency domain, it suppresses sharp spectral peaks (e.g., formants or harmonic anomalies) to reduce frequency contrast. By explicitly penalizing these residuals, the adversarial perturbation is forced to conform to a stationary texture, which the human brain naturally habituates to and cognitively ignores.

Further extended discussions are detailed in Appendix A.

## 7. Conclusion

In this paper, we propose an efficient and stealthy black-box attack method achieved through carrier selection and perturbation optimization. We achieved a 100% Attack Success Rate (SR) against Automatic Speech Recognition (ASR) systems in both the digital and physical domains. Our adversarial examples generated in the digital domain closely mimic the fluency and naturalness of benign music. Furthermore, user studies reveal that 55.6% of human participants in the physical domain were unable to distinguish our adversarial examples from benign music. Comprehensive evaluation results demonstrate that our approach outperforms state-of-the-art attack techniques in terms of both effectiveness and imperceptibility.

## Software and Data

To facilitate reproducibility and future research, we have released the source code, which are available at https://github.com/Spa-rkle/HWN-Attack.

## Acknowledgements

We thank the anonymous reviewers for their constructive comments. This work is supported in part by NSFC under Grant Nos. 62302498, 92270204, U24A20236 and the CAS Project for Young Scientists in Basic Research (Grant No. YSBR-118).

## Impact Statement

This paper presents a novel adversarial attack targeting the perceptual stealthiness of black-box ASR systems. Our research exposes a critical vulnerability where malicious commands can be concealed within music carriers, evading human attention. As voice-controlled systems (e.g., smart speakers, in-vehicle assistants) become ubiquitous, the techniques described herein could theoretically be misused to covertly manipulate devices in physical environments in a particular situation. However, we believe that highlighting this "hearing without noticing" gap is essential for the safety community. By demonstrating the vulnerability of ASR, our work motivates stricter safety standards for commercial ASR deployment and the development of more robust, low-overhead defense strategies. We strongly advocate for future defenses to incorporate multi-factor authentication (e.g., voiceprint verification) to mitigate such over-the-air threats especially in critical scenarios.

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

# A. Extended Discussion

## A.1. Potential Directions for Attack Optimization

Black-box attacks represent a formidable challenge characterized by severely restricted information access. Consequently, the volume of related research has diminished recently, suggesting a narrowing scope for optimization. Nevertheless, we envision the following avenues for future enhancement:

**Leveraging Large-Scale Carrier Datasets**   Our carrier selection algorithm enhances stealthiness by identifying segments within music that satisfy masking conditions. However, experimental results indicate that the proportion of masked frames in the selected segments does not reach 100%. This implies the existence of superior carriers within larger datasets capable of masking a greater number of frames, thereby further elevating stealthiness.

**Generative AI for Carriers or Perturbations**   Our preliminary experiments suggest that generating desired carriers or perturbations using current AI models is non-trivial, often resulting in unnatural audio artifacts or failed attacks. However, we believe that training specialized generative models to synthesize customized carriers or perturbations holds significant promise for future research.

## A.2. Potential Defense Directions

While we evaluated generic signal processing defenses like Frequency Band Filtering (Eisenhofer et al., 2021) and Audio Turbulence (Yuan et al., 2018), their effectiveness against black-box attacks is limited because such attacks often rely on robust perturbations. To provide a more comprehensive defense strategy against stealthy adversarial audio, future systems can integrate the following multi-layered approaches.

**Learning-based Acoustic Detection**   Instead of relying on heuristic filters, one potential direction is to train a dedicated classifier to detect adversarial artifacts. By leveraging learning-based acoustic forensics, the classifier can learn to distinguish the structural anomalies introduced by the attack from the natural acoustic distribution of benign music, thereby filtering out malicious audio before it is processed by the ASR model.

**Speaker Authentication and Anti-Spoofing**   Since our attack methodology utilizes TTS engines to generate the underlying command, biometric defenses offer a robust countermeasure.

- **Voice Spoofing Detection:** Systems can deploy anti-spoofing models specifically designed to identify and reject synthetic speech generated by TTS APIs, directly mitigating the threat.

- **Speaker Verification:** Many real-world ASR systems incorporate voiceprint recognition. To bypass this mechanism, an attacker would need to first acquire the victim's audio samples and employ advanced voice cloning models to synthesize the target command. This significantly escalates the cost and difficulty of physical-world attacks.

**Multi-Factor Confirmation**   At the system level, high-level semantic supervision can be applied to the ASR's text output. If the decoded command is contextually improbable or highly sensitive in the current scenario, a supervisory layer should intercept the action and require explicit user confirmation (e.g., physical interaction or a secondary prompt) before execution. In practical real-world scenarios, service providers are capable of deploying defense strategies, which inevitably incur a certain degradation in system usability. In the physical experiments, our method successfully attacked three commercial voice assistants (Gemini, Amazon Alexa, and Doubao). We are unsure whether these systems employ specific defenses. It is possible that, due to real-time constraints or concerns about degrading the recognition of benign commands, **such defenses are not deployed**.

Therefore, this work serves as a reminder to both the research community and industry that **robust defensive mechanisms are necessary** for securing ASR systems.

## A.3. Towards More Objective Metrics

The calculated Spearman correlation coefficient (Spearman, 1904) between SNR and subjective normalcy is a mere **0.511**. This weak alignment suggests that SNR is an insufficient metric for predicting the perceptual stealthiness of adversarial examples We investigated a wide range of metrics, including Weighted SNR, the energy of separated vocals, and the ASR Cross-Entropy loss. However, none of these achieved a satisfactory Spearman correlation coefficient, most of which are below 0.6. Future work could potentially leverage AI models to simulate human auditory perception, generating concrete numerical scores to evaluate the stealthiness of adversarial audio more accurately.

## A.4. Digital Success Rate and Evaluation Protocol

In the digital domain, an adversarial example is considered successful if it is recognized as the target command in a **single API query**. Since no noise or distortion is introduced during digital transmission, repeated trials are not required. **The

**success rate (SR)** is defined as the ratio of successful AEs to the total number of AEs.

Evaluating stealthiness metrics on successful AEs is intended to **ensure fairness**. Our loss function reflects the trade-off between stealthiness and attack efficiency: $L = L_{acoustic} + \lambda \times L_{stealthiness}$. When the initial optimization fails, decreasing $\lambda$ reduces stealthiness and increases the attack efficiency. Consequently, failed samples are often **"ultra-stealthy"** as their weak perturbations. Including these failed samples in the stealthiness statistics would artificially increase the stealthiness performance and lead to an unfair comparison.

### A.5. Impact of Window Length and Command Duration

In our main experiments, we empirically set the sliding window length to **3 seconds** during the carrier selection and perturbation generation phases. This duration is deliberately chosen because the target voice commands (C1–C10) typically last between 1 and 2 seconds. A 3-second window ensures that the entire command is fully encapsulated while providing sufficient surrounding temporal context for the music carrier to establish an effective auditory masking threshold. The length of this window influences the attack performance under two distinct scenarios:

**Fixed Command with Extended Window.** If the target command remains fixed , increasing the carrier window length generally enhances the perceptual stealthiness of the resulting adversarial audio. This improvement occurs because a longer window introduces a higher proportion of clean, unperturbed music relative to the adversarial segments. Since human auditory attention is heavily influenced by global acoustic context, extending the unperturbed temporal boundaries acts as a perceptual buffer. It effectively dilutes the overall saliency of the localized perturbation, making the anomaly seamlessly blend into the background music.

**Longer Commands Necessitating Extended Windows.** Conversely, if the window length must be increased to accommodate an intrinsically longer target command, the overall stealthiness and attack performance typically degrade. It is crucial to clarify that this degradation is *not* caused by the expanded window itself, but rather by the increased duration of the underlying command. A longer command inherently contains more phonetic information, necessitating a larger number of perturbed frames to manipulate the ASR model. This continuous accumulation of adversarial features inevitably overwhelms the finite masking capacity of the carrier, making the speech artifacts much easier for humans to detect.

Like previous work, we focus on 10 short commands (about four words in length) that are usually used in most real-world scenarios. For longer commands, we also conducted additional tests with one six-word command and one eight-word command. Results show that our attack remains stealthy (**Nomal Rate:68.20%, Noise Rate:9.10%, Talking Rate:22.70%, MOS:4.09, SNR:16.86**) on the command "set an alarm for six thirty". But longer command "take a picture and send it to John" means a more difficult attack, which gives users more time to react. Ensuring the success rate of attack, it is difficult to remain stealthiness (**Nomal Rate:9.10%, Noise Rate:45.50%, Talking Rate:45.50%,MOS:2.86, SNR:13.82**).

## B. Reproducibility and Comparison with Recent Baselines

Since KENKU and NI-Occam have been proven superior to Devil's Whisper (Chen et al., 2020b) (an upgraded version of CommanderSong (Yuan et al., 2018)), comparisons with the latter two are omitted. To ensure a comprehensive evaluation, we independently implemented Occam strictly following the descriptions and parameter settings in their respective papers, as their official implementations are not open-sourced. For KENKU, we utilized its open-source code to generate adversarial examples.

For more recent methods (ZQ-Attack (Fang et al., 2024) and EvilHarmony (Yuan et al., 2025)), **neither work has released its source code.** We made substantial efforts to contact the authors and reproduce them:

- For ZQ-Attack, we have not received a response to our request for the code. We also attempted an independent reimplementation based on the paper. However, we were unable to reproduce the reported performance, possibly due to the method's sensitivity to hyperparameters and the complexity of the zero-query transfer strategy.

- For EvilHarmony, we contacted the authors, who informed us that their source code is not publicly available but provided 7 sample examples. We tested these samples, but all of them failed to carry out the attacks due to recent updates in commercial ASR APIs.

Since a direct experimental comparison was not feasible despite our best efforts, we provide a detailed literature comparison in the Table 7 using the human study results for physical attacks reported in the ZQ-Attack and EvilHarmony papers. This comparison is fair and meaningful, as all three studies employ identical human perception metrics (Normal Rate, Noise Rate, and Talking Rate) with consistent definitions. As shown in the Table 7, our method outperforms ZQ-Attack across all metrics, achieving a Normal Rate that is **42.14%** higher. This aligns with expectations, as ZQ-Attack prioritizes zero-query transferability and success rates over auditory stealthiness. Furthermore, our method achieves a Normal Rate **4.57%** higher

*Table 7.* Literature comparison of physical domain performance against ZQ-attack and EvilHarmony. The best results are highlighted in **bold**.

| Method | Normal | Noise | Talking |
|---|---|---|---|
| ZQ-Attack | 13.16% | 71.58% | 15.26% |
| EvilHarmony | 51.00% | 45.00% | 5.00% |
| **Ours** | **55.57%** | **34.07%** | 10.37% |

*Table 8.* The confidence intervals (95% CI) for the Normal Rate and MOS metrics of different methods. The best results are highlighted in **bold**.

| Method | Normal | MOS |
|---|---|---|
| Occam | 14.9%±8.6% | 2.78±0.56 |
| Kenku | 39.2%±11.2% | 3.55±0.24 |
| **Ours** | **56.1%±9.9%** | **4.08±0.19** |

than that of EvilHarmony. More importantly, we evaluated the 7 official audio samples provided by EvilHarmony and found that they are no longer effective against current commercial ASR APIs due to recent system updates.

Our method's robust practical applicability against state-of-the-art systems, combined with its superior stealthiness and open reproducibility, demonstrates its distinct advantages.

## C. Human Study Statistical Details

**Sample allocation.** We recruited 200 native English-speaking participants via the Prolific platform, with ranging in age from 18 to 60 years. A total of 110 unique audio samples, including both adversarial examples and benign controls, were evaluated. Each participant listened to and rated 12 audio clips. This design ensured that each audio sample was assessed by an average of 22 independent listeners.

**Significance testing.** We conducted four significance tests: **Shapiro–Wilk test, Kruskal–Wallis H-test, Dunn's post-hoc test, and Pearson Chi-square test**. Shapiro–Wilk indicated that the data did not follow a normal distribution, justifying a non-parametric analysis framework. The Kruskal–Wallis H-test revealed a highly significant overall difference for MOS scores ($H(2) = 28.46, p = 6.6e-7 < 0.05$), while Dunn's post-hoc tests (Bonferroni adjusted) confirmed that HWN significantly outperforms both KENKU ($p = 0.007 < 0.05$) and OCCAM ($p = 3.14e-6 < 0.05$). Finally, the Pearson Chi-square test showed a significant difference for Normal Rate ($\chi^2(2) = 81.83, p = 1.7e-18 < 0.001, Cramer's V = 0.3198$), indicating that the differences between methods are statistically significant.

**Confidence intervals.** The confidence intervals (95% CI) for the Normal Rate and MOS metrics are shown in Table 8.

## D. Implementation Details and Hyperparameters

### D.1. Details of Masking-aligned Carrier Selection Algorithm

In this section, we provide the detailed parameter settings used in our Masking-aligned Carrier Selection algorithm. The implementation is based on the ISO/IEC 11172-3 (MPEG-1) psychoacoustic model 1 (Liu & Lee, 1997). All audio signals are resampled to **16 kHz**.

**Spectrogram Analysis** We utilize the Short-Time Fourier Transform (STFT) to convert time-domain signals into the frequency domain. We use a **Hann window** with an FFT size ($N_{\text{FFT}}$) of **2048** and a hop length ($L_{\text{hop}}$) of **512**, resulting in a frequency resolution of approximately 7.8 Hz and a temporal resolution of 32 ms.

**Psychoacoustic Modeling** To calculate the Global Masking Threshold, we map the linear frequency scale to the **Bark scale** using standard critical band approximations. For the specific task of hiding voice commands, we focus the masking calculation on the frequency range of **0–3000 Hz**, as this band contains the majority of phonetic information that human can hear.

**Voice Activity Detection (VAD)** To ensure accurate alignment, we ignore silent frames in the command audio. We employ the pre-trained **Silero VAD** model (Silero Team, 2025) with a confidence threshold of **0.5** to generate a binary vocal mask. Only frames classified as speech are involved in the masking score calculation.

**Segment Search Strategy** We perform a sliding window search over the carrier audio. The window length corresponds to the duration of the target command (padded to a fixed length, e.g., 3 seconds). To ensure a fine-grained search, we set a high overlap ratio of **95%** between consecutive windows. A frame is considered "masked" if at least **90%** of its frequency bins within the 0–3kHz range possess energy levels below the carrier's masking threshold.

### D.2. Hyperparameters Used in Implementation

The parameters used in our code implementation are shown in Table 9.

## E. Reconstruct Time-domain Signals

Although we obtain the adversarial magnitude spectrogram $S_m + \Delta$, the corresponding phase information is absent. Directly pairing the original carrier's phase with the modified magnitude results in **phase inconsistency**. This misalignment not only introduces audible artifacts but also prevents the reconstructed time-domain signal from accurately reproducing the target adversarial features. To resolve this, we employ the Griffin-Lim algorithm (Griffin & Lim, 1984), a standard iterative method for signal reconstruction from magnitude spectrograms.

Specifically, let the complex spectrogram of the carrier audio $m$ be denoted as $S_m e^{jP_m} = \text{STFT}(m)$, where $S_m$ is the magnitude, $P_m$ is the phase, and $j$ is the imaginary unit. We initialize the reconstruction by combining the adversarial magnitude with the original phase: $m_0 = \text{iSTFT}((S_m + \Delta)e^{jP_m})$. Subsequently, we iteratively refine the signal through two steps: ① Apply $\text{STFT}(\cdot)$ to the current estimate $m_i$ to extract its phase $P_{m_i}$; ② Enforce the adversarial magnitude constraint by replacing the magnitude of $m_i$ with $S_m + \Delta$ while retaining the updated phase, yielding the next iteration: $m_{i+1} = \text{iSTFT}((S_m + \Delta)e^{jP_{m_i}})$. This process repeats until the magnitude consistency condition is met: $\text{MSE}(|\text{STFT}(m_k)|, S_m + \Delta) < \varepsilon$, where $\varepsilon$ is a convergence threshold. The final output $m_k$ serves as the time-domain adversarial audio $m'$.

*Table 9.* Parameters used in code implementation.

| Parameter | Value | Description |
|---|---|---|
| *Audio Processing* | | |
| Sample Rate | 16000 Hz | Target sampling rate for all audio inputs |
| $N_{FFT}$ | 512 | FFT window size |
| Hop Length | 256 | STFT hop length |
| Sound dB | 75 | Target volume decibels for normalization |
| Max Amplitude | 32768.0 | Reference amplitude for 16-bit PCM |
| Clip Range | $[-1, 1]$ | Audio signal clipping range |
| *Optimization (Adam)* | | |
| Epochs | 1500 | Number of optimization iterations |
| Learning Rate | $4 \times 10^{-2}$ | Learning rate for Adam optimizer |
| Acoustic Loss Weight | 1.0 | Coefficient for acoustic loss term |
| Attention-dilution Loss Weight | $[25, 1000]$ | Coefficient for attention-dilution loss term |
| *Feature Extraction* | | |
| Mel Bands | 80 | Number of Mel filter banks |
| Square Kernel (Feature) | $[15]$ | Kernel size for feature smoothing |
| Square Kernel (Mask) | $[7]$ | Kernel size for mask generation |
| *CMA-ES Optimization* | | |
| Population Size | 15 | Number of candidates per generation |
| Sigma | 0.05 | Step size (standard deviation) |
| Max Generations | 1000 | Maximum generations for evolution |
| Early Stop | 5 generations | Stop if no improvement in 5 generations |
| *Signal Reconstruction (GLA)* | | |
| GLA Iterations | 1000 | Griffin-Lim Algorithm iterations |
| *Defense Method* | | |
| Frequency Band Filtering | $[200, 7000]$ | Frequency band used to filter signals |
| Audio Turbulence | 500 | Noise amplitude for 16-bit PCM (maximum 32768.0) |

## F. Questionnaire Used in Human Perception Study

**Human Perception Study Questionnaire**

*Instruction: Please listen to each audio sample twice and answer the following questions based on your immediate perception.*

**Q1. Perception of the Audio Content**
*After listening to the audio, which description best fits your feeling?*

☐ It is a normal sound/music; I cannot hear other anomalies.

☐ It is abnormal; I hear some background noise.

☐ It is abnormal; I hear someone speaking/whispering.

**Q2. Audio Quality Rating (MOS)**
*Compared to the reference audio, please rate the quality (1=Bad, 5=Excellent).*

○ **5** (Excellent: Imperceptible)

○ **4** (Good: Perceptible but not annoying)

○ **3** (Fair: Slightly annoying)

○ **2** (Poor: Annoying)

○ **1** (Bad: Very annoying)

**Q3. Transcription**
*If you heard speech, please write down the words:*

_______________________________________________

*(Leave blank if no speech is heard)*

**Q4. Listening Count**
*How many times did you listen before answering Q3?* _______ times.

**Q5. Tolerance**
*Would you keep listening to your favorite music if it contained such noise?*

☐ Yes

☐ No

## G. Detailed Experimental Data

We targeted the following commercial ASR systems: Google (goo, 2025), Microsoft (mic, 2025), Alibaba (ali, 2025), Tencent (ten, 2025), OpenAI (Radford et al., 2023). Attack results are shown in Tables 10, 11, and 12.

*Table 10.* Detailed performance comparison between **Microsoft** and **Tencent** ASR systems. Commands are abbreviated as C1–C10 (C1: call my wife, C2: how is the weather, C3: make it warmer, C4: navigate to my home, C5: open the door, C6: open the website, C7: play music, C8: restart phone, C9: take a picture, C10: turn off the light). All ratio values are in percentage (%).

| CMD | MICROSOFT | | | | | | | TENCENT | | | | | | |
|---|---|---|---|---|---|---|---|---|---|---|---|---|---|---|
| | NORMAL | NOISE | TALKING | 1ST | 2ND | MOS | SNR | NORMAL | NOISE | TALKING | 1ST | 2ND | MOS | SNR |
| C1 | 62.10% | 20.70% | 17.20% | 0.00% | 0.00% | 4.10 | 21.20 | 46.40% | 46.40% | 7.10% | 0.00% | 0.00% | 4.00 | 25.05 |
| C2 | 44.80% | 44.80% | 10.30% | 0.00% | 0.00% | 3.83 | 15.71 | 53.60% | 32.10% | 14.30% | 0.00% | 0.00% | 4.29 | 17.68 |
| C3 | 41.40% | 17.20% | 41.40% | 3.40% | 3.40% | 3.83 | 15.98 | 25.00% | 50.00% | 25.00% | 0.00% | 0.00% | 3.46 | 23.44 |
| C4 | 55.20% | 41.40% | 3.40% | 0.00% | 0.00% | 3.69 | 22.37 | 39.30% | 25.00% | 35.70% | 0.00% | 0.00% | 3.61 | 24.12 |
| C5 | 72.40% | 13.80% | 13.80% | 0.00% | 0.00% | 4.24 | 20.71 | 60.70% | 17.90% | 21.40% | 0.00% | 0.00% | 4.18 | 23.21 |
| C6 | 44.80% | 34.50% | 20.70% | 0.00% | 0.00% | 3.93 | 14.95 | 35.70% | 39.30% | 25.00% | 0.00% | 0.00% | 3.46 | 18.16 |
| C7 | 62.10% | 24.10% | 13.80% | 0.00% | 0.00% | 4.14 | 22.81 | 42.90% | 17.90% | 39.30% | 0.00% | 0.00% | 3.79 | 26.51 |
| C8 | 51.70% | 27.60% | 20.70% | 0.00% | 0.00% | 3.48 | 13.60 | 21.40% | 35.70% | 42.90% | 0.00% | 17.90% | 4.14 | 19.51 |
| C9 | 82.80% | 13.80% | 3.40% | 0.00% | 0.00% | 4.28 | 17.18 | 67.90% | 21.40% | 10.70% | 0.00% | 0.00% | 4.29 | 19.16 |
| C10 | 79.30% | 17.20% | 3.40% | 0.00% | 0.00% | 4.14 | 20.54 | 50.00% | 10.70% | 39.30% | 0.00% | 0.00% | 4.00 | 12.78 |
| **AVG** | **59.66%** | **25.51%** | **14.81%** | **0.34%** | **0.34%** | **3.97** | **18.51** | **44.29%** | **29.64%** | **26.07%** | **0.00%** | **1.79%** | **3.92** | **20.96** |

*Table 11.* Detailed performance evaluation on **Alibaba** and **OpenAI Whisper** ASR systems. Commands are abbreviated as C1–C10 (C1: call my wife, C2: how is the weather, C3: make it warmer, C4: navigate to my home, C5: open the door, C6: open the website, C7: play music, C8: restart phone, C9: take a picture, C10: turn off the light). All ratio values are presented in percentage (%).

| CMD | ALIBABA | | | | | | | OPENAI WHISPER | | | | | | |
|---|---|---|---|---|---|---|---|---|---|---|---|---|---|---|
| | NORMAL | NOISE | TALKING | 1ST | 2ND | MOS | SNR | NORMAL | NOISE | TALKING | 1ST | 2ND | MOS | SNR |
| C1 | 62.10% | 37.90% | 0.00% | 0.00% | 0.00% | 4.14 | 23.68 | 100.00% | 0.00% | 0.00% | 0.00% | 0.00% | 4.70 | 27.32 |
| C2 | 82.80% | 10.30% | 6.90% | 0.00% | 0.00% | 4.66 | 17.93 | 50.00% | 43.30% | 6.70% | 0.00% | 0.00% | 3.97 | 21.57 |
| C3 | 69.00% | 20.70% | 10.30% | 0.00% | 0.00% | 4.38 | 20.31 | 73.30% | 26.70% | 0.00% | 0.00% | 0.00% | 4.50 | 21.89 |
| C4 | 31.00% | 37.90% | 31.00% | 0.00% | 0.00% | 3.86 | 25.11 | 63.30% | 26.70% | 10.00% | 0.00% | 0.00% | 4.27 | 26.03 |
| C5 | 58.60% | 31.00% | 10.30% | 0.00% | 3.40% | 4.21 | 21.20 | 86.70% | 10.00% | 3.30% | 0.00% | 0.00% | 4.60 | 30.27 |
| C6 | 24.10% | 34.50% | 41.40% | 0.00% | 0.00% | 3.41 | 19.44 | 30.00% | 43.30% | 26.70% | 0.00% | 0.00% | 3.50 | 22.20 |
| C7 | 24.10% | 48.30% | 27.60% | 0.00% | 0.00% | 3.74 | 21.37 | 36.70% | 43.30% | 20.00% | 0.00% | 0.00% | 3.63 | 28.62 |
| C8 | 6.90% | 37.90% | 55.20% | 0.00% | 0.00% | 2.83 | 15.56 | 23.30% | 53.30% | 23.30% | 0.00% | 0.00% | 3.50 | 25.34 |
| C9 | 65.50% | 20.70% | 13.80% | 0.00% | 0.00% | 4.45 | 20.88 | 63.30% | 33.30% | 3.30% | 0.00% | 0.00% | 4.10 | 25.03 |
| C10 | 17.20% | 48.30% | 34.50% | 3.40% | 6.90% | 3.79 | 23.75 | 30.00% | 53.30% | 16.70% | 0.00% | 0.00% | 3.80 | 25.78 |
| **AVG** | **44.13%** | **32.75%** | **23.10%** | **0.34%** | **1.03%** | **3.95** | **20.92** | **55.66%** | **33.32%** | **11.00%** | **0.00%** | **0.00%** | **4.06** | **25.40** |

*Table 12.* Detailed digital comparison results. Commands are abbreviated as C1–C10 (C1: call my wife, C2: how is the weather, C3: make it warmer, C4: navigate to my home, C5: open the door, C6: open the website, C7: play music, C8: restart phone, C9: take a picture, C10: turn off the light). Metrics: Perception (Nor: Normal, Noi: Noise, Tlk: Talking, 1st, 2nd), and Quality (MOS, SNR). All ratio values are in percentage (%).

| CMD | OURS | | | | | | | KENKU | | | | | | | OCCAM | | | | | | |
|---|---|---|---|---|---|---|---|---|---|---|---|---|---|---|---|---|---|---|---|---|---|
| | NOR | NOI | TLK | 1ST | 2ND | MOS | SNR | NOR | NOI | TLK | 1ST | 2ND | MOS | SNR | NOR | NOI | TLK | 1ST | 2ND | MOS | SNR |
| C1 | 66.3% | 26.7% | 7.0% | 0.0% | 0.0% | 4.2 | 24.4 | 18.5% | 14.8% | 66.7% | 7.4% | 11.1% | 3.5 | 15.6 | 23.3% | 23.3% | 53.3% | 0.0% | 0.0% | 3.3 | 7.9 |
| C2 | 65.5% | 26.8% | 7.6% | 0.0% | 0.0% | 4.3 | 19.0 | 22.2% | 3.7% | 74.1% | 11.1% | 11.1% | 3.5 | 12.5 | 6.7% | 26.7% | 66.7% | 0.0% | 0.0% | 2.2 | 4.4 |
| C3 | 60.3% | 23.6% | 16.1% | 0.7% | 0.7% | 4.2 | 21.1 | 25.9% | 7.4% | 66.7% | 0.0% | 0.0% | 3.2 | 12.3 | 37.9% | 17.2% | 44.8% | 0.0% | 0.0% | 4.0 | 10.7 |
| C4 | 50.6% | 31.9% | 17.4% | 0.0% | 0.0% | 3.9 | 24.2 | 48.1% | 33.3% | 18.5% | 0.0% | 0.0% | 3.7 | 18.8 | 0.0% | 27.6% | 72.4% | 0.0% | 0.0% | 2.7 | 8.1 |
| C5 | 73.5% | 14.5% | 11.9% | 0.0% | 0.7% | 4.4 | 24.0 | 85.2% | 14.8% | 0.0% | 0.0% | 0.0% | 3.9 | 19.9 | 29.6% | 37.0% | 33.3% | 0.0% | 3.7% | 3.6 | 12.1 |
| C6 | 34.8% | 40.3% | 24.9% | 0.0% | 0.0% | 3.6 | 18.4 | 48.1% | 14.8% | 37.0% | 0.0% | 0.0% | 3.9 | 15.6 | 7.4% | 7.4% | 85.2% | 3.7% | 11.1% | 2.9 | 5.5 |
| C7 | 48.2% | 31.0% | 20.9% | 0.0% | 0.0% | 3.9 | 24.9 | 37.0% | 11.1% | 51.9% | 3.7% | 14.8% | 3.8 | 15.2 | 21.4% | 39.3% | 39.3% | 0.0% | 0.0% | 3.1 | 12.0 |
| C8 | 36.4% | 34.5% | 29.1% | 0.0% | 3.6% | 3.7 | 18.3 | 29.6% | 3.7% | 66.7% | 11.1% | 3.7% | 3.7 | 12.3 | 7.1% | 21.4% | 71.4% | 0.0% | 0.0% | 2.9 | 10.9 |
| C9 | 74.5% | 19.3% | 6.2% | 0.0% | 0.0% | 4.4 | 20.6 | 63.0% | 29.6% | 7.4% | 3.7% | 0.0% | 4.1 | 17.2 | 35.7% | 46.4% | 17.9% | 0.0% | 0.0% | 3.8 | 9.0 |
| C10 | 52.4% | 28.0% | 19.5% | 0.7% | 1.4% | 4.0 | 20.0 | 29.6% | 7.4% | 63.0% | 11.1% | 11.1% | 3.8 | 17.1 | 25.0% | 21.4% | 53.6% | 0.0% | 3.6% | 3.7 | 15.8 |
| **AVG** | **56.3%** | **27.7%** | **16.1%** | **0.1%** | **0.6%** | **4.1** | **21.5** | **40.7%** | **14.1%** | **45.2%** | **4.8%** | **5.2%** | **3.7** | **15.7** | **19.4%** | **26.8%** | **53.8%** | **0.4%** | **1.8%** | **3.2** | **9.6** |

