# OpenReview forum: "Hearing Without Noticing? Attention-Aware Stealthy Black-Box Adversarial Audio Attacks"
_ICML.cc/2026/Conference — ICML 2026 regular_

### Official Review · Reviewer_Dj5F · 2026-03-06

**Soundness:** 3
**Presentation:** 4
**Significance:** 3
**Originality:** 4
**Overall Recommendation:** 4
**Confidence:** 4

**Summary:**

The paper proposes a novel method for generating stealthy black-box adversarial audio attacks against Automatic Speech Recognition (ASR) systems. Instead of solely minimizing the magnitude of adversarial perturbations, the authors focus on reducing the auditory attention these perturbations attract to achieve cognitive stealthiness. This is accomplished through two main components: 1) a masking-aligned carrier selection algorithm based on the MPEG-1 psychoacoustic model, which identifies music segments that naturally mask the target command's frequencies, and 2) an attention-diluted optimization strategy that utilizes Structural-Residual Decomposition on the log-Mel spectrogram to suppress salient, attention-grabbing spectral peaks. The attacks are evaluated against 5 commercial cloud APIs and 3 physical voice assistants. Through a user study with 200 participants, the authors demonstrate the approach achieves a 100% physical attack success rate while being perceived as benign music by 55.6% of participants, outperforming state-of-the-art baselines like KENKU and Occam.

**Compliance With Llm Reviewing Policy:**

Affirmed.

**Final Justification:**

The rebuttal addresses most of my concerns. My remaining concern is the justification of the attention-dilution loss, the rebuttal improves the intuition and cites relevant auditory-saliency literature, but the formulation still appears closer to a well-motivated heuristic regularizer than a rigorously validated auditory model.

**Key Questions For Authors:**

See the weaknesses section for the key questions.

**Limitations:**

yes

**Strengths And Weaknesses:**

Strengths:
1. Shifting the focus from standard magnitude constraints to modeling human cognitive selective attention via psychoacoustic masking and spectral smoothness is an insightful and highly relevant direction for audio adversarial examples. The application of structural-residual decomposition to audio stealthiness is a creative adaptation.
2. The empirical evaluation is extensive and realistic. Testing against multiple commercial APIs and widely used virtual assistants demonstrates real-world applicability. The large-scale human perception study with detailed perception breakdowns is a very strong point that convincingly supports the stealthiness claims.
3. The method effectively improves perceptual stealthiness over strong baselines while maintaining a perfect 100% physical attack success rate.

Weaknesses:
1. For the digital black-box API attacks, the method uses CMA-ES to optimize the perturbation. CMA-ES typically requires a large number of iterative queries (Table 6 suggests a population of 15 and up to 1000 generations). The threat model explicitly states attackers are constrained by a "limited, acceptable budget of paid queries" (Section 3.2), yet the paper omits the average number of queries required to craft a successful adversarial example. Stating the process takes "less than 1 hour" (Section 5.7) is insufficient, as it does not reflect the exact query cost or the likelihood of triggering API rate limits.
2. The "Attention-Dilution Loss" relies on Structural-Residual Decomposition (Hou & Zhang, 2007), a visual saliency technique designed for 2D natural images. Applying 2D spatial smoothing convolutions over a log-Mel spectrogram treats the frequency and time dimensions uniformly and symmetrically. While this effectively acts as a smoothness regularizer empirically, its specific formulation lacks rigorous justification from an auditory psychoacoustics perspective.
3. Table 5 shows that the attack's success rate for stealthy samples drops significantly under basic defenses like Frequency Band Filtering and Audio Turbulence. This implies the crafted perturbations are somewhat fragile and might fail in noisy physical environments or actively defended systems.

---

> ### Author Rebuttal · Authors · 2026-03-31
>
> We sincerely appreciate the reviewer's professional insights.
>
> ---
> ### 1. **Query Efficiency in Digital Black-box Attacks**
> We tested the **query budgets** in the digital settings (**Zero** in the physical domain):
> - In the digital-domain experiments, CMA-ES refinement averaged *47* iterations and *707* queries per sample (*30000* queries in Occam), taking *778* seconds on average. The average cost for an adversarial audio across differect commercial APIs is *$0.43*. The average query speed is *0.91* query per second which doesn't trigger any API rate limits.
>
> We will include detailed query budgets in the revision.
>
> ---
> ### 2. **Auditory Justification for Structural-Residual Decomposition**
> We agree that the Attention-Dilution Loss functions similarly to a smoothness regularizer. From the perspective of psychoacoustics, auditory saliency (Kayser et al., 2005) identifies **frequency contrast** and **temporal contrast** as evidence features for auditory attention. The 2D spatial smoothing operation therefore serves as a unified proxy to suppress these two primary triggers of auditory attention.
> - **In the time domain**, it smooths sudden transients and impulsive clicks, reducing temporal contrast.
> - **In the frequency domain**, it suppresses sharp spectral peaks such as formants or harmonic anomalies, reducing frequency contrast.
> By enforcing these constraints, the adversarial perturbation is encouraged to conform to a stationary texture, which is more likely to be cognitively ignored according to auditory saliency theory.
>
> In addition, the results in Table 4 of the ablation study indicate that the attention-dilution loss plays a critical role in stealthiness, as removing it leads to a *46.29%* decline in Normal Rate.
>
> ---
> ### 3. **Robustness and Fragility under Defenses**
> It is indeed common that defenses such as frequency filtering or audio turbulence can significantly disrupt the attack performance of adversarial audios. However, as shown in Table 5, our method still demonstrates **stronger robustness** than the baselines on stealthy samples.
>
> In the physical experiments, our method also **successfully attacked three commercial voice assistants** (*Gemini, Amazon Alexa, and Doubao*). We are unsure whether these systems employ specific defenses. It is possible that, due to real-time constraints or concerns about degrading the recognition of benign commands, **such defenses are not deployed**.
>
> Therefore, this work serves as a reminder to both the research community and industry that **robust defensive mechanisms are necessary** for securing ASR systems.

---

> > ### Author Rebuttal · Reviewer_Dj5F · 2026-04-02
> >
> > The rebuttal addresses most of my concerns. My remaining concern is the justification of the attention-dilution loss, the rebuttal improves the intuition and cites relevant auditory-saliency literature, but the formulation still appears closer to a well-motivated heuristic regularizer than a rigorously validated auditory model.

---

> > > ### Author Response · Authors · 2026-04-04
> > >
> > > Thank you for the thoughtful comment. We have benefited from your feedback. We agree that our formulation is a psychoacoustically based regularizer, and we will state this clearly in the revision. However, modeling stealthiness from the perspective of subjective auditory perception remains a clear challenge, and we are glad to have taken a step in this direction with promising empirical results. Thank you again for your time and effort in reviewing our paper.

---

### Official Review · Reviewer_wdqx · 2026-03-09

**Soundness:** 3
**Presentation:** 2
**Significance:** 3
**Originality:** 3
**Overall Recommendation:** 5
**Confidence:** 3

**Summary:**

This paper proposes an algorithm for generating stealthy targeted audio adversarial examples by embedding a spoken command into a music carrier such that commercial ASR systems transcribe the target command while humans primarily perceive music. The method is motivated by selective auditory attention and combines two main components: 1) masking-aligned carrier selection, which scores candidate carrier segments via a recall-like masking coverage metric and selects the best segment, and 2) perturbation shaping, which optimizes an acoustic loss in the mel-spectrogram domain together with an attention-dilution regularizer in the STFT domain. The final waveform is reconstructed either via Griffin-Lim (phase recovery) or via amplitude scaling with a CMA-ES-based post-processing step. Experiments cover 10 commands across five commercial ASR APIs (digital domain) and three voice assistants (physical domain), along with a 200-participant user study to evaluate perceptual stealthiness.

**Compliance With Llm Reviewing Policy:**

Affirmed.

**Final Justification:**

My questions are well addressed, specifically about Q3 and Q4, and it changed my evaluation accordingly.

**Key Questions For Authors:**

Key Questions for Authors
- Digital-domain SR claim: The conclusion states 100% SR in both the digital and physical domains. Where is the digital-domain SR reported, and under what definition/protocol?
- Why is the carrier-selection window length $L_{win}$ fixed? Intuitively it may need to depend on command length or phonetic content. Did you test adaptive window sizes, and does performance change?
- Have you tried using raw audio waveform rather than spectrogram space? Modern models, such as XLS-R or WavLM, take raw signals as input, and what would be a way to generate adversarial audio?
- What genres or characteristics of carriers are most frequently selected as optimal across different commands? Can you provide the selected carriers (or genre distribution) and how it correlates with success/stealthiness?
- In Table 3, the results appear to vary substantially across commands. Do you have analysis explaining why some commands are significantly easier/harder?

**Limitations:**

The approach assumes access to a sufficiently large and diverse pool of music carrier candidates. In practice, curating/collecting suitable carriers may require nontrivial human effort and domain knowledge, which could limit scalability or deployment in constrained settings.

**Strengths And Weaknesses:**

## Strengths
- The overall design choices are well aligned with the paper’s stated motivation around auditory masking / selective attention.
- The evaluation is broad in terms of platforms: multiple commercial ASR APIs and multiple physical voice assistants, plus a relatively large-scale user study (200 participants) targeting the “without noticing” goal.
- Ablation study is informative to understand their structure of algorithm.
- Responsible discussion of societal impact / mitigations: The paper includes an explicit discussion of potential defense directions and mitigation strategies for the malicious use of the proposed attack in the Appendix, which improves the responsible framing of the work.

## Weaknesses
- For digital attack results, the authors only evaluate successful adversarial examples, which might bias the stealthiness. This way of evaluation does not mirror how successful this algorithm is, and how bad it could be when they fail to generate adversarial examples.
- The report for success rate can be more helpful if it uses metrics such as “1/3, 2/3, 3/3” success rates.
- The paper discusses the acoustic context factor for adversarial attack generation, but it does not give us information about the genres of candidate music carriers. It would strengthen soundness to test across broader genres/tempos levels.


## Minor issues / typos
- Even though the paper mentioned about ‘perturbation shaping’, it could be assumed that $\Delta$ may be fixed since it contains the command. In Section 4.2, it is worth to mention again about the optimization variables for clarity.
- Section 4.2.2. first sentence: “steathiness” -> “stealthiness”
- Use consistent loss naming between the main body (acoustics loss and attention-dilution loss) and appendix (feature loss and smoothness loss)
- Notation: in Section 4.3, consider using a different symbol for the amplitude scaling parameters (since $\lambda$ is already used as a regularization weight in Section 4.2)
- Tables: add “better direction” arrows consistently (e.g., Noise, SR) or explicitly justify why not.

---

> ### Author Rebuttal · Authors · 2026-03-31
>
> We sincerely appreciate the reviewer's professional insights.
>
> ---
> ### 1. **Digital Success Rate and Evaluation Protocol**
> In the digital domain, an adversarial example is considered successful if it is recognized as the target command in a **single API query**. Since no noise or distortion is introduced during digital transmission, repeated trials are not required. **The success rate (SR)** is defined as the ratio of successful AEs to the total number of AEs. We will add this clarification in the revision and report SR in Table 1.
>
> Evaluating stealthiness metrics on successful AEs is intended to **ensure fairness**. Our loss function reflects the trade-off between stealthiness and attack efficiency: $L=L_{\text{acoustic}}+ \lambda \times L_{\text{stealthiness}}$. When the initial optimization fails, decreasing $\lambda$ reduces stealthiness and increases the attack efficiency. Consequently, failed samples are often **``ultra-stealthy''** as their weak perturbations. Including these failed samples in the stealthiness statistics would artificially increase the stealthiness performance and lead to an unfair comparison.
>
> ---
> ### 2. **Fixed Window Length**
> In our experiments, the window size is *3 seconds*, as it **covers** all commands (C1–C10), which typically last *1–2 seconds*. Window length can affect performance in different ways.
> - For a fixed command, **increasing the window length may enhance perceptual stealthiness**, as more of the segment can contain clean music and less contains adversarial perturbations.
> - However, for longer commands that require extending the window to cover them, overall performance may decrease (refer to Q2 to Reviewer 2NRM). This decrease is due to the increased command length rather than the window length itself.
>
> We will conduct additional human studies on the impact of window length and discuss the results in the revision.
>
> ---
> ### 3. **Raw Audio Waveform vs. Spectrogram Space**
> In the early stages of our research, we indeed extensively explored the feasibility of direct perturbation on raw waveforms. We ultimately chose the **spectrogram domain** based on two considerations:
> - Our core modules, such as the **masking-aligned carrier selection** and **attention dilution loss**, are inherently frequency-domain based. Although STFT is differentiable, the magnitude extraction discards phase information, leading to a highly non-convex loss landscape that causes severe **gradient oscillation** and **slow convergence** during direct waveform optimization.
> - Unlike regular speech, our music background contains highly coordinated harmonic structures. Direct perturbation of the raw waveform disrupts the delicate phase relationships across frequencies, inevitably **introducing audible broad-band noise and artifacts**, thereby compromising the attack's stealthiness.
>
> ---
> ### 4. **Characteristics of the Selected Carriers**
> Based on our observations, we conclude the following two characteristics:
> - Optimal carriers tend to exhibit **high frequency density and high energy** (*82.72%* of all carriers). This is reasonable, as quiet and sparse music segments are more likely to provide insufficient coverage to conceal the command's phonemic features.
> - Optimal carriers are more likely to have **dynamic and complex rhythms** (*70.91%* of all carriers) compared to steady beats. We hypothesize that such rhythms better match varied and dynamic commands.
>
> A discussion of carrier characteristics will be added in the Appendix.
>
> ---
> ### 5. **Analysis of Command Variance**
> We analyzed **the number of voiceless and voiced sounds** in the commands, as well as **the distribution of long vowels**, but did not observe a consistent pattern. We suspect this may be related to the carrier dataset or the frequency of the commands.
> - For example, *``take a picture"* or *``open the door"* may be more common in ASR applications than ``*open the website*," making them easier for ASR models to recognize.
>
> We will conduct further analysis and include it in the revision if we obtain significant findings.
>
> ---
> ### 6. **Minor Issues**
> We will address them and proofread the entire paper.

---

> > ### Author Rebuttal · Reviewer_wdqx · 2026-04-01
> >
> > I appreciate the authors for the reply! My questions are all resolved, and I'll adjust the score accordingly.

---

> > > ### Author Response · Authors · 2026-04-02
> > >
> > > Thank you sincerely for your acknowledge of our work. Your insights are beneficial to our work. Thanks again for your time and effort in reviewing our paper.

---

### Official Review · Reviewer_TWc7 · 2026-03-11

**Soundness:** 2
**Presentation:** 3
**Significance:** 3
**Originality:** 3
**Overall Recommendation:** 4
**Confidence:** 4

**Summary:**

This paper studies stealthy adversarial audio attacks against black-box ASR systems, with a particular focus on over-the-air physical-world settings. The method improves stealthiness through two components: carrier selection and an attention-dilution loss. Experiments are conducted in both digital and physical-world settings, with additional human studies and ablation analysis.

**Compliance With Llm Reviewing Policy:**

Affirmed.

**Final Justification:**

My concerns have been largely addressed, so I raised the score.

**Key Questions For Authors:**

Could the authors include comparisons with SOTA baselines, or at least explain why these recent baselines were not considered? It would also strengthen the paper to provide more complete statistical reporting for the human study, as well as clearer validation that the proposed attention-aware approach correlates with human perception.

**Limitations:**

Yes

**Strengths And Weaknesses:**

Strengths
- Improving stealthiness from an attention-aware perspective is reasonably novel.
- The empirical results are strong, especially in the physical-world setting.

Weaknesses
- The central “attention-aware” claim is still not fully substantiated. The proposed formulation appears more like an effective heuristic regularizer than a well-validated proxy for human auditory attention. The paper does not provide sufficient evidence that the proposed surrogate consistently correlates with actual human perceptual attention.
- Baselines are outdated, which weakens the empirical claims. The compared methods seem to stop at around 2023, while more recent and relevant methods, such as ZQ-Attack(2024)[1] and EvilHarmony(2025)[2], are not included. As a result, the current experiments mainly demonstrate improvements over older methods, rather than convincingly establishing superiority over the current state of the art.
- Human study is not reported in enough detail. Although the paper includes user studies, the main paper lacks important statistical details such as sample allocation, significance testing, and confidence intervals. These details are important given that the paper’s main claim is about being less noticeable to human listeners.
- The generalization and threat-model transparency are limited. The experiments mainly rely on a specific set of music carriers, which makes it unclear whether the method generalizes to broader audio carrier types. In addition, the reporting of black-box query cost, prior information, and optimization budget is not sufficiently detailed.

[1]F. Zheng, et al. "Zero-query adversarial attack on black-box automatic speech recognition systems." Proceedings of the 2024 on ACM SIGSAC Conference on Computer and Communications Security. 2024.
[2]X. Yuan et al., "EvilHarmony: Stealthy Adversarial Attacks Against Black-Box Speech Recognition Systems," 2025 IEEE Symposium on Security and Privacy (SP).

---

> ### Author Rebuttal · Authors · 2026-03-31
>
> We sincerely appreciate the reviewer's professional insights.
>
> ---
> ### 1.**The ``Attention-Aware" Claim and Theoretical Substantiation**
> - Our attention-dilution loss is grounded in established psychoacoustics. The **auditory saliency work** (Kayser et al., 2005) demonstrates through human detection experiments that **frequency contrast** and **temporal contrast** are evidence features for auditory attention. Following this theory, we aim to reduce both frequency and temporal contrast in the perturbation to improve stealthiness.
> - To achieve this, we adapt **Structural-Residual Decomposition** by performing 2D spatial smoothing convolutions over the log-Mel spectrogram. This smoothing regularizer reduces frequency and temporal contrast during optimization. **In the time domain**, it smooths sudden transients and impulsive clicks, reducing temporal contrast. **In the frequency domain**, it suppresses sharp spectral peaks such as formants or harmonic anomalies, reducing frequency contrast.
>
> We will clarify this further in the revision.
>
> ---
> ### 2. **Comparisons with ZQ-Attack and EvilHarmony**
> **Neither work has released its source code.** We made substantial efforts to contact the authors and reproduce them:
> - For ZQ-Attack, we have **not yet received a response** to our request for the code. We also attempted an independent reimplementation based on the paper. However, we were unable to reproduce the reported performance, possibly due to **the method’s sensitivity to hyperparameters and the complexity of the zero-query transfer strategy**. Notably, the results reported in their paper (*Normal Rate: 13.16%, Noise Rate: 71.58%, Talking Rate: 15.26%*) are *consistently lower* than ours (***Normal Rate: 55.57%, Noise Rate : 34.07%, Talking Rate :10.37%***) across all metrics.
> - For EvilHarmony, we contacted the authors, who informed us that their source code is **not publicly available but provided 7 sample examples**. We tested these samples, but ***all of them failed*** to carry out the attacks due to recent updates in commercial ASR APIs.
>
> Since a direct experimental comparison was not feasible despite our best efforts, we will include a detailed literature comparison in the Appendix.
>
> ---
> ### 3. **Human Study Statistical Details**
> Human study statistical data details:
> - **Sample allocation**. We recruited *200* native English-speaking participants via the Prolific platform. A total of *110* unique audio samples, including both adversarial examples and benign controls, were evaluated. Each participant listened to and rated *12* audio clips. This design ensured that each audio sample was assessed by an average of *22* independent listeners.
> - **Significance testing**. We conducted four significance tests: *Shapiro–Wilk test*, *Kruskal–Wallis H-test*, *Dunn’s post-hoc test*, and *Pearson Chi-square test*. Shapiro–Wilk indicated that the data did not follow a normal distribution, justifying a non-parametric analysis framework. The Kruskal–Wallis H-test revealed a highly significant overall difference for MOS scores (***H(2) = 28.46, p = 6.6e-7 < 0.05***), while Dunn’s post-hoc tests (Bonferroni adjusted) confirmed that HWN significantly outperforms both KENKU (***p = 0.007 < 0.05***) and OCCAM (***p = 3.14e-6 < 0.05***). Finally, the Pearson Chi-square test showed a significant difference for Normal Rate (***χ²(2) = 81.83, p = 1.7e-18 < 0.001, Cramer’s V = 0.3198***), indicating that the differences between methods are statistically significant.
> - **Confidence intervals**. The confidence intervals (***95% CI***) for the Normal Rate and MOS metrics are as follows:
>
> |        | Normal       | MOS         |
> | ------ | ------------ | ----------- |
> | Our    | 56.1%±9.9%     | 4.08±0.19   |
> | Kenku  | 39.2%±11.2%    | 3.55±0.24   |
> | Occam  | 14.9%±8.6%     | 2.78±0.56   |
>
> We will include these comprehensive results in the Appendix to ensure full transparency and reproducibility.
>
> ---
> ### 4. **Generalization and Threat-model Transparency**
> > Results for different carrier types (e.g., ambient noise, piano) are detailed in our reply to ***Q1 of Reviewer 2NRM***.
>
> We analyze the query cost, prior information requirements, and optimization budget under both digital and physical black-box settings.
> - **Query Cost**. In the digital-domain experiments, CMA-ES refinement averaged *47* iterations and *707* queries per sample (*30000* queries in Occam), taking *778* seconds on average. The average cost for an adversarial audio across differect commercial APIs is *$0.43*. In the physical experiments, ***no interactive queries*** could be applied to the target black-box devices, resulting in zero queries to the target system.
> - **Prior Information**. We only need an acceptable budget of paid queries in the digital domain and ***no prior information*** in the physical domain.
> - **Optimization Budget**. Details of optimization budget are shown in Table 6.
>
> We will include detailed query costs in the revision.

---

> > ### Author Rebuttal · Reviewer_TWc7 · 2026-04-03
> >
> > Thank you for the detailed rebuttal and clarification.
> >
> > My main concern remains the strength of the experimental comparison. Without direct comparison to stronger and more recent baselines, it is still difficult to convincingly demonstrate the advantage of the proposed approach.
> >
> > Since this issue was not resolved in the rebuttal, I maintain my original score.

---

> > > ### Author Response · Authors · 2026-04-06
> > >
> > > We appreciate the reviewer's valid concern. To address this concern, we provide a comparison using the human study results for physical attacks reported in the ZQ-Attack and EvilHarmony papers. This comparison is fair and meaningful, as all three studies employ identical human perception metrics (*Normal Rate, Noise Rate, and Talking Rate*) with consistent definitions. The comparison results are as follows.
> > >
> > > | Method | Normal Rate (%) | Noise Rate (%) | Talking Rate (%) |
> > > | :--- | :--- | :--- | :--- |
> > > | ZQ-Attack | 13.16 | 71.58 | 15.26 |
> > > | EvilHarmony | 51.00 | 45.00 | 5.00 |
> > > | Ours | **55.57** | 34.07 | 10.37 |
> > >
> > > -	As shown in the comparison, our method outperforms ZQ-Attack across all metrics, achieving a Normal Rate that is ***42.14%*** higher. This aligns with expectations, as ZQ-Attack prioritizes zero-query transferability and success rates over auditory stealthiness. Furthermore, our method achieves a Normal Rate ***4.57%*** higher than that of EvilHarmony.
> > > -	More importantly, we evaluated the *7* official audio samples provided by EvilHarmony and found that they are ***no longer effective*** against current commercial ASR APIs due to recent system updates.
> > > -	We believe that our method's robust practical applicability against **state-of-the-art systems**, combined with its **superior stealthiness** and **open reproducibility**, demonstrates its distinct advantages.
> > >
> > > We sincerely appreciate your time and effort in reviewing our work.

---

### Official Review · Reviewer_2NRM · 2026-03-13

**Soundness:** 3
**Presentation:** 3
**Significance:** 2
**Originality:** 4
**Overall Recommendation:** 4
**Confidence:** 4

**Summary:**

This paper proposes Hearing Without Noticing (HWN), a stealthy black-box adversarial audio attack against ASR systems. Instead of only minimizing perturbation magnitude, the method aims to reduce human auditory attention to the perturbation. It combines masking-aligned carrier selection to find music segments with strong masking capacity and an attention-dilution loss to suppress attention-sensitive spectral residuals while matching the target command’s Log-Mel features. Experiments on five commercial ASR APIs and three voice assistants show improved stealthiness over prior black-box baselines while maintaining strong attack success.

**Compliance With Llm Reviewing Policy:**

Affirmed.

**Final Justification:**

My concerns have been addressed, so I am maintaining my positive score.

**Key Questions For Authors:**

1. How does the method perform on sparser or non-musical backgrounds such as piano, ambient sounds, or white noise?

2. How do attack success and stealthiness change for substantially longer commands? Does the method require long contiguous carrier segments with consistently high masking thresholds in such cases?

3. Have the authors tested different TTS voices, different TTS engines, or real human recordings as target templates?

**Limitations:**

yes

**Strengths And Weaknesses:**

Strengths

1. Clear motivation and problem framing

The paper is motivated by a sensible observation: perturbation magnitude alone is not a sufficient proxy for perceptual stealthiness, since human auditory perception is attention-driven. This perspective is intuitive and well connected to psychoacoustic masking and auditory saliency.

2. Strong empirical results

The method shows clear improvements over prior black-box baselines on both attack success and stealthiness metrics, including human-study-based evaluations. The ablation study also suggests that the main components make a meaningful contribution to performance.

3. Human perception evaluation adds value

The paper includes a human study with 200 participants and reports perceptual categories such as Normal, Noise, and Talking, along with MOS. This is valuable given that the main claim of the paper is improved stealthiness against human listeners.

Weaknesses

1. Limited carrier-type generalization

The experiments use only 27 electronic and rhythmic music tracks as candidate carriers. Since these carriers likely have dense spectra and strong masking capacity, it remains unclear whether the method would remain effective for sparser or structurally different backgrounds, such as piano, ambient audio, or white noise.

2. Target commands are short, and longer-command scenarios are not analyzed

The evaluation uses only 10 short, formulaic assistant-style commands. In addition, the carrier-selection window is tied to target-command duration, and masking is computed frame by frame over vocal frames, so longer commands may require sufficiently long carrier segments with consistently strong masking ability. The paper does not analyze this setting.

3. Limited analysis of TTS-template generalization

The method uses a single Google TTS system to synthesize the target command audio and optimizes the adversarial example toward that template. It remains unclear how sensitive the method is to different TTS voices, different speaker styles, or real human recordings.

4. Query efficiency is not reported transparently

Although the paper emphasizes limited query budgets and includes a CMA-ES-based refinement step in the digital setting, it does not clearly report the actual query count, iteration count, or per-sample query cost. This makes the efficiency claim difficult to assess in practice.

---

> ### Author Rebuttal · Authors · 2026-03-31
>
> We sincerely appreciate the reviewer's professional insights.
>
> ---
> ### 1. **Limited Carrier-type Generalization**
> To verify the generalization of our method with sparser carriers like piano, we conducted additional tests using **sparse piano music and ambient sound**：
>
> | Method | SR | Normal | Noise | Talk | MOS | SNR |
> | :--- | :--- | :--- | :--- | :--- | :--- | :--- |
> | Ours | 100% | 36.35% | 59.10% | 4.50% | 4.05 | 14.91 |
> | Kenku | 100% | 18.15% | 22.75% | 59.10% | 2.14 | 6.87 |
> | Occam | 70% | 13.60% | 54.50% | 31.80% | 2.32 | 4.69 |
>
> - The results indicate that the method could generalize to sparser carriers, achieving an *100%* success rate while maintaining ***much better stealthiness*** than other methods, with *Normal Rate: 36.35%,, Noise Rate: 59.10%, Talking Rate: 4.50%, MOS: 4.05, and SNR: 14.91*.
> - We also observe that using rhythmic music as carriers achieves **better stealthiness** compared to sparse piano music, which is reasonable since rhythmic music has higher global masking threshold curve.
>
> ---
> ### 2. **Target Command Length and Analysis**
> Like previous work, we focus on *10* short commands (about four words in length) that are usually used in most real-world scenarios.
> - For longer commands, we also conducted additional tests with one six-word command and one eight-word command:
> - Results show that our attack remains stealthy (***Nomal Rate:68.20%, Noise Rate:9.10%, Talking Rate:22.70%, MOS:4.09, SNR:16.86***) on the command *``set an alarm for six thirty"*.
> - But longer command *``take a picture and send it to John"* means a more difficult attack, which gives users more time to react. Ensuring the success rate of attack, it is difficult to remain stealthiness (*Nomal Rate:9.10%, Noise Rate:45.50%, Talking Rate:45.50%,MOS:2.86, SNR:13.82*).
>
> ---
> ### 3. **TTS-template Generalization**
> We evaluated a subset of commands using different TTS service (Microsoft Azure Speech) and speaker styles:
>
> | Voice Type | SR | Normal | Noise | Talk | MOS | SNR |
> | :--- | :--- | :--- | :--- | :--- | :--- | :--- |
> | Man with Low Voice | 100% | 36.40% | 45.50% | 45.50% | 3.82 | 15.28 |
> | Woman with Happy Voice | 100% | 77.30% | 4.50% | 18.20% | 4.36 | 18.18 |
> | Human Recordings | 100% |  50% | 13.6% | 36.4% | 3.95 | 13.13 |
> | Baseline (Woman of Google TTS) | 100% | 55.57% | 34.07% | 10.37% | 4.19 | 16.12 |
>
> - Compared with the baseline, our method exhibits ***robustness across different TTS services and voice styles***. Notably, we achieved exceptionally high imperceptibility (***Normat Rate 77.30%***) when optimizing toward a ``Woman with happy voice" template.
> - We did observe a relative decrease in performance when using a ``man with low voice." The distinct low-frequency characteristics of a male voice are more challenging to mask under relative high-frequency background.
> - Human recordings perform slightly worse, showing an *5.57%* decline in Normal Rate, possibly due to device distortion.
>
> ---
> ### 4. **Query Efficiency Transparency**
> We tested the **query budgets** in the digital settings (**Zero** in the physical domain):
> - In the digital-domain experiments, CMA-ES refinement averaged *47* iterations and *707* queries per sample (*30000* queries in Occam), taking *778* seconds on average. The average cost for an adversarial audio across differect commercial APIs is *$0.43*.
>
> We will include detailed query budgets in the revision.

---

> > ### Author Rebuttal · Reviewer_2NRM · 2026-04-04
> >
> > My concerns have been addressed, so I am maintaining my positive score.

---

> > > ### Author Response · Authors · 2026-04-05
> > >
> > > We are glad that our response could address your concerns. Your insights are valuable to our work. Thank you sincerely for your time and effort in reviewing our paper.

---

### Decision · Program_Chairs · 2026-04-30

**Decision:**

Accept (regular)

**Comment:**

The paper studies stealthy adversarial audio attacks against black-box ASR systems. The attack consists in embedding a spoken command into a music carrier such that commercial ASR systems transcribe the target command while humans primarily perceive music.

There is a clear trade-off between stealthiness and attack success and the paper addresses it in a reasonably novel approach. The empirical evaluation is strong, involving 200 participants to evaluate stealthiness. Attacks are conducted on 5 cloud API and 3 voice assistants.

During the rebuttal the following weaknesses were reported and not fully addressed:
1. the competitors are limited to quite old methods. Unfortunately, the code of the recent competitors is unavailable, even though the authors requested it. Hence we think that it cannot be a limitation.
2. The attention-dilution loss is based on a technique for 2D images but is applied to log-Mel spectrograms. This has not been justified. The deployment in real settings can be altered by noisy or real environments. I do not consider these two shortcomings to be grounds for rejection.
3. The proposed defenses are generic and limited to Frequency Band Filtering and Audio Turbulence. Voice spoofing, speaker verification methods could be applied or one can train a classifier for this task.

The authors correctly answered several other points that are not described in this meta review. At the end of the rebuttal phase all reviewers are positive on this paper.